# Increased Susceptibility of *Rousettus aegyptiacus* Bats to Respiratory SARS-CoV-2 Challenge Despite Its Distinct Tropism for Gut Epithelia in Bats

**DOI:** 10.3390/v16111717

**Published:** 2024-10-31

**Authors:** Björn-Patrick Mohl, Claudia Blaurock, Angele Breithaupt, Alexander Riek, John R. Speakman, Catherine Hambly, Marcel Bokelmann, Gang Pei, Balal Sadeghi, Anca Dorhoi, Anne Balkema-Buschmann

**Affiliations:** 1Institute of Novel and Emerging Infectious Diseases, Friedrich-Loeffler-Institut, Federal Research Institute for Animal Health, Suedufer 10, 17493 Greifswald-Insel Riems, Germany; bjoern-patrick.mohl@fli.de (B.-P.M.); claudia.blaurock@unibe.ch (C.B.); bokelmannm@rki.de (M.B.); balal.sadeghi@fli.de (B.S.); 2Department of Experimental Animal Facilities and Biorisk Management, Friedrich-Loeffler-Institut, Federal Research Institute for Animal Health, Suedufer 10, 17493 Greifswald-Insel Riems, Germany; angele.breithaupt@fli.de; 3Institute of Animal Welfare and Animal Husbandry, Friedrich-Loeffler-Institut, Federal Research Institute for Animal Health, Doernbergstraße 25, 29223 Celle, Germany; alexander.riek@fli.de; 4School of Biological Sciences, University of Aberdeen, Aberdeen AB24 2TZ, UK; j.speakman@abdn.ac.uk (J.R.S.); c.hambly@abdn.ac.uk (C.H.); 5Centre for Biological Threats and Special Pathogens, Robert Koch Institute, Nordufer 20, 13353 Berlin, Germany; 6Institute of Immunology, Friedrich-Loeffler-Institut, Federal Research Institute for Animal Health, Suedufer 10, 17493 Greifswald-Insel Riems, Germany; gang.pei@fli.de (G.P.); anca.dorhoi@fli.de (A.D.)

**Keywords:** SARS-CoV-2, *Rousettus aegyptiacus*, tissue tropism, respiratory tract, digestive tract, viral infection

## Abstract

Increasing evidence suggests bats are the ancestral hosts of the majority of coronaviruses. In general, coronaviruses primarily target the gastrointestinal system, while some strains, especially Betacoronaviruses with the most relevant representatives SARS-CoV, MERS-CoV, and SARS-CoV-2, also cause severe respiratory disease in humans and other mammals. We previously reported the susceptibility of *Rousettus aegyptiacus* (Egyptian fruit bats) to intranasal SARS-CoV-2 infection. Here, we compared their permissiveness to an oral infection versus respiratory challenge (intranasal or orotracheal) by assessing virus shedding, host immune responses, tissue-specific pathology, and physiological parameters. While respiratory challenge with a moderate infection dose of 1 × 10^4^ TCID_50_ caused a systemic infection with oral and nasal shedding of replication-competent virus, the oral challenge only induced nasal shedding of low levels of viral RNA. Even after a challenge with a higher infection dose of 1 × 10^6^ TCID_50_, no replication-competent virus was detectable in any of the samples of the orally challenged bats. We postulate that SARS-CoV-2 is inactivated by HCl and digested by pepsin in the stomach of *R. aegyptiacus*, thereby decreasing the efficiency of an oral infection. Therefore, fecal shedding of RNA seems to depend on systemic dissemination upon respiratory infection. These findings may influence our general understanding of the pathophysiology of coronavirus infections in bats.

## 1. Introduction

Since late 2019, severe acute respiratory syndrome coronavirus-2 (SARS-CoV-2) has caused more than 775 million human infections with over 7 million fatalities (https://ourworldindata.org/covid-cases (accessed on 12 August 2024)). While the virus predominantly replicates within the respiratory tract in humans, SARS-CoV-2 can also colonize the gastrointestinal tract (GIT) and may cause severe damage [1,2,3]. Although the origin of SARS-CoV-2 is still not fully elucidated [4,5,6], closely related Betacoronavirus sequences, have repeatedly been detected in bats, underlining a possible role of bats as reservoir hosts for SARS-CoV-2, with the potential risk of a spillover into the human population [7,8].

Bats have recurrently been reported as reservoir hosts for zoonotic viruses [9], such as lyssaviruses, including rabies virus [10], Marburg virus [11], and henipaviruses [12]. Interestingly, bats were not known to harbor coronaviruses (CoV) before the SARS-CoV epidemic in 2003, presumably due to limited surveillance in bats [13,14,15]. Since then, an increasing number of CoVs have been detected in numerous bat species globally, including the Americas [16], Africa [17], Asia [18], Australasia [19,20], and Europe [21]. Bats have even been postulated as an ancestral host for a number of Alpha- and Betacoronaviruses that are correlated to other mammalian hosts today [22]. In bat field samples, fecal specimens or anal swabs have yielded most of the successful detections of Alpha- and Beta-CoV, indicating a possible preferential tropism for the GIT in these species [13,23,24,25,26]. Although, while CoV RNA has also been detected in oral swab field samples [27,28,29], paired fecal samples yielded longer read lengths and coverage during sequencing [28] or were more often RNA positive compared to oral swabs [29]. A tropism of a Beta-CoV for intestinal epithelium and the underlying connective tissues in the insectivorous bat *Pipistrellus nathusii* has been recently reported [30]. *Rhinolophus* bats are the reservoir hosts for the diversity of Sarbecoviruses (*Sarbecovirus* subgenus within the *Betacoronavirus* genus) that includes SARS-CoV [31], and they have also been postulated as a reservoir host for SARS-CoV-2 [7]. Furthermore, these bats may also facilitate CoV co-infections and recombination events between CoV strains from different geographical locations [32]. Egyptian fruit bats (*R. aegyptiacus*), which were used in this study, belong to the large family of *Pteropodidae*, whose members have been reported to harbor a great diversity of viruses, including species from the families *Coronaviridae*, *Paramyxoviridae*, *Reoviridae* and *Filoviridae*, including Marburg virus [24,33].

We previously reported that *R. aegyptiacus* fruit bats were susceptible to intranasal SARS-CoV-2 infection and transiently supported the virus propagation in their upper respiratory tract but showed no signs of disease and did not efficiently transmit the virus to contact animals [34]. Based on these results, we questioned the role of the lower respiratory tract and the GIT in SARS-CoV-2 infections in this species. We, therefore, compared the efficiency of intranasal, orotracheal, and oral infection routes by assessing physiological parameters, virus shedding, host immune responses and pathological changes. Since we had shown virus propagation only in the nasal and tracheal epithelium upon intranasal challenge [34], the orotracheal route was utilized to investigate whether the lower respiratory tract was at all permissive to SARS-CoV-2 infection, as the inoculation route affects the site of initial virus replication [35]. Among many other CoVs, both SARS-CoV [31], as well as its close relative RaTG13 [7], were detected in fecal swabs of *Rhinolophus* bats, and SARS-CoV-2 has been shown to be able to infect *Rhinolophus sinicus*-derived intestinal organoids [36]. These findings indicate a GIT tissue tropism of these viruses in bats, which prompted us to also include the oral infection route in this study. It is, however, unclear whether the GIT infection occurred directly following ingestion or resulted from systemic dissemination. Since the presence and distribution of cellular receptors have a significant role in SARS-CoV-2 tropism [37], and the epithelium of the small intestine has been shown to express the cellular receptor ACE2 [38] as well as the protease TMPRSS2 [39] in *Rhinolophus* bats, the latter seems highly plausible. However, intestinal organoids from *Rousettus leschenaultii* and *Artibeus jamaicensis* turned out to be refractory to SARS-CoV-2 [39,40]. To better understand the fate of SARS-CoV-2 virions in the GIT of *R. aegyptiacus*, we first determined the pH in the different sections of the GIT, as these data were not sufficiently available from the literature [41,42]. We also analyzed the stability of SARS-CoV-2 virions in an acidic environment and their susceptibility to digestion by pepsin.

Furthermore, we analyzed the immune responses in different tissues of the infected and uninfected bats. Interferons (IFNs), as key cytokines controlling the host’s antiviral state, are categorized into three distinct groups, designated as type I, type II, and type III IFNs [43,44]. In most mammals, type I IFNs, including IFNα and IFNβ, are produced by diverse cell types upon viral infection [43]. However, it was shown that in the black flying fox *Pteropus alecto*, IFNα is constitutively expressed, even in the absence of viral infection [45]. We also quantified the expression level of IFNγ, a type II IFN that is crucial for host defense against intracellular pathogens and mainly produced by lymphocytes. Similar to type I and II IFN, type III IFN can restrict SARS-CoV-2 replication [46] and induce lung tissue damage during SARS-CoV-2 infection [47]. Thus, we also evaluated type III IFN expression in *R. aegyptiacus* bats in this study.

In addition to the classical virological, immunological, and histopathological analyses, our study included the monitoring of physiological parameters, including body weight as well as body temperature and locomotion activity, through intraperitoneally implanted data loggers. Moreover, we measured the energy expenditure using the doubly labeled water (DLW) method, as described previously [48,49]. We consider this broad approach crucial since bats, as wildlife animals, often do not display clear clinical signs, making the interpretation of experimental infections challenging.

This approach allowed us to thoroughly assess the SARS-CoV-2 infection in *R. aegyptiacus* bats and to shed light on the tissue tropism of SARS-CoV-2 in this species, especially in the gastrointestinal tract. This approach can be translated to other CoV infections in bats. A better understanding of the roles of the different infection and transmission routes will be helpful for assessing and managing possible spillover events of other zoonotic viruses from bats to other mammals.

## 2. Materials and Methods

All infectious work was performed in the Friedrich-Loeffler-Institut (FLI) biosafety level 3 (BSL-3) laboratory and animal facility, Insel Riems, Germany.

### 2.1. Cells, Viruses and Data Availability

Vero E6 cells were obtained from the FLI’s cell-culture collection in Veterinary Medicine. The SARS-CoV-2 isolate 2019_nCoV Muc-IMB-1 (accession number LR824570 [50]) was kindly provided by the Bundeswehr Institute of Microbiology, Munich, Germany. The complete genome is available in GISAID: EPI_ISL_406862. The virus was propagated and maintained in Vero E6 cells in DMEM supplemented with 2% FCS. Prior to its use in animal studies, the virus sequence was verified as described [51] with a few modifications, as reported previously [52].

### 2.2. Virus Titration

Titration of virus stocks and samples collected during this study (swab samples, nasal lavage samples, tissues) were performed using 80–90% confluent Vero E6 cells in 96-well plates (Corning, Kennebunk, ME, USA) and tissue culture infectious dose 50 (TCID_50_) was determined as described before [52]. Titers were calculated using the Spearman and Kaerber method [53].

### 2.3. R. aegyptiacus Challenge Experiments

Male and female fruit bats from the *R. aegyptiacus* breeding colony at the FLI were randomly assigned to the study groups of six animals. Animals were housed in groups of three in cages of 75 × 130 × 65 cm. All animals had *ad libitum* access to water and fresh fruits. Clinical scores (posture, behavior, food intake) were monitored daily, whereas body weight could only be measured when bats were under short isoflurane inhalation anesthesia for the collection of nasal lavage samples every other day. Handling and sampling were always performed, starting with the uninfected group, to minimize the contamination risk.

### 2.4. Inoculation Routes and Sampling

In the first experiment, six bats per group were inoculated with 1 × 10^4^ TCID_50_ (in the following termed as low dose) in a 100 µL volume by the oral, orotracheal, or intranasal route, the latter two under short isoflurane anesthesia, and were then monitored for 14 days. Three uninfected animals were kept under identical conditions and sampled together with the infected animals. Oral swab samples were collected one day before infection and at 1, 3, 5, 7, and 11 days post-inoculation (dpi) in 500 µL minimum essential medium (MEM) containing Penicillin/Streptomycin (P/S) (Millipore Sigma, Darmstadt, Germany). Nasal lavage samples were collected at 2, 4, 6, 8, and 14 dpi by flushing 200 µL of PBS along both nostrils. Anal swab samples were collected daily from −1 dpi to 8 dpi, as well as 11 dpi. Samples were stored at −80 °C until further analysis. At 4 and 14 dpi, three animals per group were euthanized by deep isoflurane anesthesia followed by cardiac exsanguination and cervical dislocation. Organ samples (nose, trachea, lung, lung lymph node, colon, small intestine, mesenteric lymph node, brain) were collected and immediately frozen for further RT-qPCR analysis as well as for virus titration. Blood from each bat was collected, and serum was stored at −20 °C until further analysis was performed (Appendix A).

In a second experiment, six *R. aegyptiacus* bats per group were inoculated either via the oral route or the intranasal route with 1 × 10^6^ TCID_50_ (in the following termed as high dose). The experimental setup was the same as described for the first round, but this time, necropsies were performed at 2 and 6 dpi (Appendix A).

### 2.5. Determination of Physiological Parameters

#### 2.5.1. Body Core Temperature and Locomotor Activity

The body core temperature and locomotor activity were measured for two animals per group in both studies using an intraperitoneally implanted data logger DST micro-ACT (Star Oddi, Gardabaer, Iceland). In the second experiment, data loggers DST micro-ACT and DST nano-T (Star-Oddi, Gardabaer, Iceland) measuring the body core temperature were implanted in an additional animal per group. The DST micro-ACT was programmed to record temperature and acceleration-based activity every 5 min from 00:00 to 12:00 and every 10 min from 12:00 to 00:00 throughout the study. The installation of the data loggers, data retrieval, and interpretation were performed as previously described [54].

#### 2.5.2. Daily Energy Expenditure (DEE)

The DEE was determined individually for 29 bats (first trial: three uninfected, four orally infected, four orotracheally infected, four intranasally infected; second trial: three uninfected, five orally infected, six intranasally infected) for the initial 48 h post-infection, using the DLW method [55,56]. On the first and last day of the DEE measurements, the precise body mass for each bat was recorded. Initially, a blood sample was drawn from each analyzed bat by puncturing the uropatagial vein for the determination of the background isotopic enrichment of 2H and 18O in the body fluids [57]. Subsequently, each bat was injected intraperitoneally with 2.76 ± 0.05 g DLW per kg body mass (65% 18O and 35% 2H; 99.90% purity). All bats were then kept without access to food or water for an equilibration period of 1 h, after which a further blood sample was taken, and the animals were given access to food and water again. At 48 h post-DLW administration, a final blood sample was taken to determine the isotope elimination rates. All blood samples were stored at −20 °C until determination of 18O and 2H levels [55,56]. Analysis of the isotopic enrichment of blood was performed blind, using a Liquid Isotope Water Analyzer (Los Gatos Research, Mountain View, CA, USA) [58]. Samples were run alongside five lab standards for each isotope and International standards to correct delta values to ppm. A single-pool model was used to calculate rates of CO_2_ production as recommended for use in animals less than 5 kg in body mass [59].

### 2.6. Virological and Pathohistological Analysis

#### 2.6.1. RNA Extraction and qRT-PCR for the Quantification of SARS-CoV-2 Viral RNA

Total RNA was extracted from oral and anal swab samples, nasal lavage, and tissue samples using the ‘Viral RNA/DNA isolation NucleoMag^®^VET kit’ (Macherey & Nagel GmbH, Düren, Germany) in a KingFisher Flex Purification System (ThermoFisher Scientific, Waltham, MA, USA) as published previously [52]. SARS-CoV-2 RNA was detected using a published qRT-PCR protocol [34,60]. Viral genome copy numbers were calculated from standard curves determined for 10^−2^ to 10^−5^ dilutions containing known copy numbers of SARS-CoV-2. All samples were also analyzed for the presence of subgenomic RNA (sgRNA) as an indication of virus replication [61,62].

#### 2.6.2. Histopathology, Immunohistochemistry, RNA In Situ Hybridization

A full necropsy was performed on all animals, and samples from the nasal conchae, trachea, oesophagus, lung, spleen, liver, heart, kidney, stomach, small intestine, large intestine, and brain were fixed in 10% neutral buffered formalin. Tissues were paraffin-embedded, and 2–3 μm sections were stained with hematoxylin and eosin (HE). For SARS-CoV-2 antigen detection, a monoclonal antibody against the SARS-CoV nucleocapsid protein (clone 4F3C4, diluted 1:50 [63]) was used on consecutive sections as described [34]. As a negative control, consecutive sections were labeled with an irrelevant antibody (M protein of Influenza A virus, ATCC clone HB-64). A positive control slide from a SARS-CoV-2-infected Syrian hamster was included in each run.

RNA in situ hybridization (RNA ISH) for the detection of SARS-CoV-2 RNA was performed on selected tissues for verification of positive PCR results (brain sample FH25/2 dpi; intestinal tissue FH35/6 dpi). The RNAScope™ 2-5 HD Reagent Kit-Red (ACD, Advanced Cell Diagnostics, Newark, CA, USA) was applied according to the manufacturer’s instructions. For RNA-ISH, RNAScope™ probes were custom-designed for the nucleocapsid protein. As technical assay controls, a positive control probe (*peptidylprolyl isomerase B*) and a negative control probe (*dihydrodipicolinate reductase*) were included. All sides were scanned using a Hamamatsu S60 scanner, and an evaluation was performed using the NDPview.2 plus software (Version 2.8.24, Hamamatsu Photonics, K.K., Hamamatsu City, Japan) by a board-certified pathologist (AB, DiplECVP) in a blinded fashion.

Macroscopic lesions were recorded, and HE-stained sections of all tissues were evaluated and described. Following immunohistochemistry (IHC) and ISH, the distribution of virus antigen or RNA was graded on an ordinal scale with scores 0 = no detection, 1 = focal, affected cells/tissue <5% or up to 3 foci per tissue; 2 = multifocal, 6–40% affected; 3 = coalescing, 41–80% affected; 4 = diffuse, >80% affected. The target cell was identified based on the morphology.

### 2.7. Analysis of Early and Late Immune Response

#### qRT-PCR for the Quantification of Immune Gene Expression and for the Expression of Cellular Receptor ACE2 as Well as the Protease TMPRSS2

One µg purified RNA was subsequently utilized for cDNA synthesis with the LunaScript RT SuperMix Kit (New England BioLabs, Ipswich, MA, USA). The qRT-PCR reactions were carried out with Luna^®^ Universal qPCR Master Mix (New England BioLabs, Ipswich, MA, USA) according to manufacturers’ instructions. The reaction setup was as follows: 95 °C for 2 min; (95 °C for 30 s, 62 °C for 30 s, 72 °C for 1 min) for 40 cycles; 72 °C for 10 min and infinite hold at 4 °C. Unless stated otherwise, each qPCR reaction was performed with 100 ng cDNA. To minimize pipetting errors, the template was diluted, and 5 µL was used for each qRT-PCR reaction. Measurements were performed with the QuantStudioTM 6 Flex Real-Time PCR System (Applied Biosystems, Waltham, MA, USA). The average threshold cycle (Ct) of quadruplicate reactions was employed for all subsequent calculations using the ΔCt method. Gene expression was normalized to eukaryotic translation elongation factor 1 alpha 1 (EEF1A1), and fold changes were calculated against corresponding controls [64].

We also quantified the *ACE2* and *TMPRSS2* expression levels in the analyzed tissues of three mock-infected animals using the PCR protocol described above.

The primer sequences used in this study are shown in Table 1.

### 2.8. Serological Analyses

All serum samples were analyzed using a microsphere-based assay, and all results were confirmed by indirect ELISA.

#### 2.8.1. Microsphere-Based Assay

For this, the receptor binding subdomain 1 of spike protein (RBD-SD1) of 2019_nCoV Muc-IMB-1 was coupled to magnetic COOH beads using the BioPlex Coupling Kit, using the buffers included in the kit (Biorad, Munich, Germany), following the manufacturer’s instructions. Next, 0.5 µL beads in 50 µL bead buffer per sample were added to the wells of a black 96-well plate (Greiner, Frickenhausen, Germany). The bead buffer was removed by placing the plate on a magnetic plate for 2 min and carefully aspirating the buffer. After addition of 50 µL of serum samples diluted 1:100 in PBS-T to the wells, the plate was incubated on a shaker at 850 rpm for 1 h at room temperature (RT). Next, the plate was washed three times with PBS-T by placing it on a magnetic plate for 2 min and again aspirating the wash buffer before adding 50µL of a mouse anti-human IgG Fc clone HP6017 (BioLegend, San Diego, CA, USA), showing a broad reactivity to mammalian IgG including bat IgG, diluted 1:200 per well, and the plate was incubated on a shaker at 850 rpm for 1 h at RT. Then, the plate was again washed three times with PBS-T before 50 µL of an Alexa Fluor 532 goat anti-mouse IgG H+L (Thermo Fisher Scientific, Waltham, MA, USA) in a dilution of 1:100 was added per well, and the plate was incubated on a shaker at 850 rpm for 1 h at RT. Finally, the plate was washed three times with PBS-T by placing it on a magnetic plate for 2 min and aspirating the wash buffer before 125 µL of Sheath buffer was added per well, and the plate was incubated on a shaker at 850 rpm for 2 min at RT. Fluorescence was measured in a Bio-Plex200 reader (Biorad, Munich, Germany). The FI values determined for the analyzed samples were set in relation to the FI value of our positive control (serologically positive *R. aegyptiacus* serum samples collected during an earlier study [34]) and are displayed as PP (percent of the positive control). The cutoff value for this assay was calculated as 9.0 PP, based on the results determined for the serum samples collected prior to the infection and the serum samples of the mock-infected bats (*n* = 42).

#### 2.8.2. Indirect SARS-CoV-2 RBD ELISA

SARS-CoV-2 specific antibodies were detected using a published protocol with a few modifications, including the use of Protein A/G in a dilution of 1:30,000 [52,65]. The optical density (OD) values determined for the analyzed samples were set in relation to the OD value of our positive control (a serologically positive *R. aegyptiacus* serum sample collected during an earlier study [34]) and are displayed as PP (percent of the positive control). The cutoff value for this assay was calculated as 20.94 PP, based on the results determined for the serum samples collected prior to the infection and the serum samples of the mock-infected bats (*n* = 42).

### 2.9. SARS-CoV-2 Sensitivity to pH and Pepsin

#### 2.9.1. pH Measurement of Different Sections of the GIT

The pH of GIT tissue samples from the stomach, duodenum, jejunum, ileum, colon, and rectum of 4 uninfected healthy animals was measured using pH-measurement stripes (MQuant pH-indicator strips Universal indicator—Supelco, Merck KGAa, Darmstadt, Germany) by swabbing the exposed epithelial lining.

#### 2.9.2. Stability of SARS-CoV-2 Proteins Under Acidic Conditions and in Homogenates from Stomach and Small Intestine Samples

We prepared 10% (*w*/*v*) tissue homogenates of *R. aegyptiacus* stomach and small intestine samples in homogenization buffer (cOmplete™ Protease Inhibitor buffer, Roche, Basel, Switzerland) using a tissuelyser (UPHO ultimate homogenizer, Geneye). This protease inhibitor buffer specifically inhibits serine and cysteine proteases but does not affect metallo- and aspartic proteases. 10^8,63^ TCID_50_ SARS-CoV-2 in 100 µL was mixed with 150 µL 1% homogenate per reaction and mixed gently. We then added 3.5 µL 1 M HCl (final pH of 2.5), or an equal volume of H_2_O, to the corresponding controls. Following the addition of these components, the reactions were gently mixed and then incubated at 37 °C for 1 and 5 min, respectively. The pH of each sample was neutralized using 1.5 M Tris pH 8.8.

Next, 10^8,63^ TCID_50_ SARS-CoV-2 in 100 µL was mixed with 150 µL H_2_O before 3.5 µL of 1 M HCl was added to this mixture (final pH of 2.5), and the corresponding control was treated with an equal volume of H_2_O. Following the addition of HCl or H_2_O, the reaction volumes were gently mixed and incubated at 37 °C, and a sample was collected after 1 and 5 min. The sample pH was neutralized via the addition of 1.5M Tris pH 8.8. Virus titrations were then performed as described above.

#### 2.9.3. Sensitivity of SARS-CoV-2 Proteins to Pepsin Proteolysis

To confirm that pepsin was involved in the proteolysis in stomach homogenate samples, we prepared 10% (*w*/*v*) homogenates of *R. aegyptiacus* stomach samples in homogenization buffer (cOmplete™ Protease Inhibitor buffer (Roche)) containing 1 mM EDTA. Pepstatin A (40 µM final concentration) (Roche) was added to two aliquots of 200 µL tissue homogenate each, and an equal volume of undiluted DMSO (Roth) carrier vehicle was added to two additional aliquots, mixed gently and then incubated at 37 °C for 10 min. Next, 125 µL of 10^4,5^ TCID_50_/mL SARS-CoV-2 were added to each reaction and gently mixed. 1 M HCl was added to one sample treated with Pepstatin A and one treated with DMSO. An equal volume of H_2_O was added to the corresponding controls. These samples were gently mixed and incubated at 37 °C for 1, 5, and 10 min, respectively. Sample pH was neutralized using 1.5 M Tris pH 8.8.

#### 2.9.4. Western Blot Analysis for Quantification of Pepsin Sensitivity of SARS-CoV-2

SDS-PAGE loading buffer was added to the samples from the stability studies before incubation at 95 °C for 10 min and loaded onto SDS-PAGE gels. SDS-PAGE gels were transferred to PVDF transfer membranes and blocked for 4 h with PBS containing 0.05% Tween 20 (PBS-T) containing 10% (*w*/*v*) skimmed milk powder. The primary antibodies that were used for the detection of Pepsin ((200-1176-0100) (Rockland, Limerick, PA, USA), 1:5000), GAPDH ((MA5-15738) (Invitrogen, Waltham, MA, USA), 1:5000), SARS-CoV-2 nucleo (N)-protein (BSV-N-05 (BioServ, Raritan Township, NJ, USA), 1:1000), and SARS-CoV-2 spike (S)-protein (BSV-S2-01(BioServ), 1:1000), were added to membranes in PBS-T containing 5% (*w*/*v*) skimmed milk powder and incubated overnight at RT. Membranes were washed with PBS-T, and then incubated with secondary antibodies Goat-anti-mouse-HRP ((Dianova, Hamburg, Germany), 1:2,000), and Donkey-anti-goat-HRP ((Dianova, Germany), 1:10,000) in PBS-T containing 5% (*w*/*v*) skimmed milk powder for 1 h. Western blots were developed via chemiluminescence using ECL (Clarity Western ECL substrate (170-5061) (RIO-RAD, South Granville, NSW, Canada)).

SDS-PAGE gels were incubated with Protein Ark Quick Coomassie Stain (Protein Ark, Rotherham, UK) for 2 h and washed in distilled H_2_O to destain and remove background staining. Coomassie blue stained SDS-PAGE gels and immunoblots were analyzed via semi-quantified densitometry using ImageJ 1.54g (Rasband, W.S., ImageJ, U. S. National Institutes of Health, Bethesda, MD, USA) [66].

### 2.10. Statistical Analysis

Mean values determined for the experimental groups were compared using unpaired *t*-test and analysis of variance (ANOVA) with Tukey’s post-hoc tests for multiple comparisons. Before performing these tests, the data distribution and normality were assessed to ensure appropriate statistical analysis. The normality of the data was evaluated using the Shapiro–Wilk test and the homogeneity of variances was checked using Levene’s test. Given that the data met the assumptions of normality, parametric tests were selected. Data was analyzed using SPSS software (IBM Corp. Released 2011. IBM SPSS Statistics for Windows, Version 20.0, IBM Corporation, Armonk, NY, 230 USA), with a significance level set at *p*-value < 0.05.

Data from the DLW study were analyzed using the aov (ANOVA) function in RStudio version 1.4.1103 [67]. We also compared the relation between body mass and DEE in bats with published DEE values in bat species measured by the DLW method (see Appendix A). For that purpose, we assessed our results with published data on DEE and body mass in bat species using the phylogenetic general least square (PGLS) approach in order to account for the potential lack of independence between species because of their shared evolutionary history. The statistical procedures have been described in detail elsewhere [68,69,70,71]. The phylogeny was derived from a published mammalian supertree, which includes 4510 species with updated branch lengths derived from dated estimates of divergence times [72]. The supertree for mammals was pruned to include only the species of concern, i.e., bats (*n* = 11), using the ‘Analysis in phylogenetics and evolution’ package [73] and the ‘Analysis of evolutionary diversification’ package [74] in RStudio. The method of PGLS was implemented for the log-transformed trait data using the ‘Comparative analyses of phylogenetics and evolution’ package [75] in RStudio using Pagel’s branch length transformation (lambda, λ), determined by maximum likelihood [76].

### 2.11. Exclusion of One Animal

One animal challenged orally with the high dose (ID FH 35) displayed results that fully matched those of the intranasally challenged bats. We, therefore, cannot exclude that a proportion of the inoculum was accidentally instilled intranasally in this animal, and this animal was, therefore, completely excluded from the analysis.

## 3. Results

### 3.1. SARS-CoV-2 Cellular Receptor ACE2 as Well as the Protease TMPRSS2 Are Expressed in Analyzed Tissues

To understand the ability of SARS-CoV-2 to establish infection in different tissues, the gene expression levels of ACE2 and TMPRSS2, which are critical for SARS-CoV-2 cell entry, were evaluated by qPCR. Except for the trachea, the analyzed tissues from the respiratory and digestive tract exhibited high expression levels of ACE2, while their expression in the brain and spleen was lower. Surprisingly, the lung lymph node showed the highest expression of ACE2 among all tested tissues. TMPRSS2 expression in these tissues was also easily detectable, albeit at lower levels (Figure 1). Altogether, both ACE2 and TMPRSS2 are expressed in the respiratory and digestive tract of *R. aegyptiacus*.

### 3.2. SARS-CoV-2 Challenge Induces Minimal Changes in Physiological Parameters

We did not observe any persistent clinical signs of disease in any of the challenged animals, such as altered body posture or reduced food intake.

During the necropsy, early-stage pregnancy was noticed in two of the uninfected bats, two intranasally infected bats and one bat each that were challenged with the low dose by the orotracheal and the oral route. In the high-dose study, one of the orally challenged bats was identified as pregnant during the necropsy.

#### 3.2.1. No Infection-Related Change in Body Weight upon SARS-CoV-2 Challenge

In the first study using the low dose, orally and orotracheally challenged bats showed some weight loss until 2 dpi, while the weight of intranasally infected bats remained unchanged during this time period. From 4 dpi until the end of the experiment, all animals gained weight (Appendix A).

In the second study using a high dose, all animals steadily increased their body weight during the 6-day period (Appendix A). The weight development of the pregnant animals did not differ from that of the other animals in this study.

#### 3.2.2. Minor Changes in Locomotor Activity and Body Core Temperature upon SARS-CoV-2 Challenge

We predominantly observed a nocturnal circadian rhythm, with body temperature and activity levels being highest during the night in both experiments (Figure 2 and Figure 3, Appendix A). Handling of the animals during the application of DLW, virus challenge, and regular sampling during the day caused an increase in body temperature combined with low levels of locomotor activity (Figure 2C and Figure 3C). However, due to low animal numbers per group, we could not conduct a statistical analysis of these physiological data.

In the low-dose challenge experiment, the uninfected control continued a clearly synchronous nocturnal circadian rhythm, while the challenged animals partly lost this rhythm within the first three days post-infection (Figure 2A). All challenged bats displayed a marginally higher mean body temperature and locomotor activity profile than the uninfected bats during the day (Figure 2B,D, Appendix A). While the intranasally challenged bats displayed values close to those of the uninfected controls, the values determined for the orotracheally and orally challenged bats were distinctly higher during this period (Appendix A). However, these divergent mean body temperature and locomotor activity values were already observable at the start of measurements at −1 dpi (Figure 2A,C). Interestingly, the orotracheally and even the orally challenged bats showed a larger variability ranging between 33.5 and 40.6 °C and 32.6 and 40.7 °C, respectively, while the intranasally challenged bats displayed body temperature values between 35.4 and 40.3 °C (Figure 2B). During the night, all challenged bats displayed a lower locomotor activity compared to the uninfected controls, with the orotracheally and orally challenged bats displaying the lowest activity (Figure 2D, Appendix A). Conversely, during the day, the orotracheally and orally challenged bats displayed greater activity than the uninfected controls (Figure 2D, Appendix A).

In the high-dose challenge experiment, all animals continued a synchronous nocturnal circadian rhythm. All groups showed a similar body temperature profile, with a lower mean body temperature during the day and a higher temperature during the night (Figure 3A). Moreover, the mean body temperature profiles were also similar for all groups (Figure 3B, Appendix A). The orally challenged group displayed a slightly higher locomotor activity level during the day, while during the night, all challenged bats showed slightly lower locomotor activity levels than the uninfected control (Figure 3D, Appendix A). As before, orally challenged bats displayed a lower locomotor activity during the night phase between 2 and 4 dpi (Figure 3C) whilst displaying a higher body temperature during this phase (Figure 3A). Intranasally challenged bats displayed a lower mean locomotor activity, while orally challenged bats displayed a similar profile as the uninfected bats (Figure 3D). Due to the small number of animals carrying these data loggers per group, a statistical analysis of the obtained data was not feasible.

#### 3.2.3. Unaltered Daily Energy Expenditure in Challenged Bats

We observed no difference in the cumulated DEE (F_3,25_ = 0.05; *p* = 0.985) and total water intake (TWI) (F_3,25_ = 0.28; *p* = 0.837) in the animals infected by different inoculation routes using both SARS-CoV-2 infection doses (Appendix A; F_3,26_ = 0.03, *p* = 0.98). The DEE of all bats (both dose groups) averaged 214 ± 7 kJ/d for uninfected bats and 206 ± 17 kJ/d, 208 ± 3 kJ/d, and 213 ± 38 kJ/d for orally, intranasally and orotracheally SARS-CoV-2 infected bats, respectively. The average for all DEE measurements was 209 ± 8 kJ/d. Similarly to DEE, TWI was not affected by the infection route (Appendix A; F_3,25_ = 0.28; *p* = 0.837). Total water intake averaged 105 ± 4 mL/d for uninfected bats and 110 ± 13 mL/d, 101 ± 5 mL/d, and 112 ± 22 mL/d for all orally, intranasally and orotracheally SARS-CoV-2 infected bats, respectively. The average for all TWI measurements was 105 ± 5 mL/d (Appendix A). Published DEE values of 11 bat species, ranging in body mass from 7.3 g (*Pipistrellus pipistrellus*) to 80.8 g (*Phyllostomus hastatus*), were available for phylogenetic analysis in correlation to our results on *R. aegyptiacus* bats (98–128 g). The resulting regression equation was DEE (kJ d^−1^) = 5.79 body mass^0.75±0.12^ with an estimated maximum likelihood λ of 0 (Appendix A), i.e., the equation followed an ordinary least square regression.

### 3.3. Virus Shedding and Tissue Distribution Following SARS-CoV-2 Challenge

#### 3.3.1. Virus Shedding Following Oral Infection Is Lower than After Intranasal or Orotracheal Challenge

First, we analyzed the susceptibility of *R. aegyptiacus* bats to inoculation with 10^4^ TCID_50_ SARS-CoV-2 using three different infection routes (intranasal, orotracheal, and oral). We were able to detect both viral genomic RNA (gRNA) and subgenomic RNA (sgRNA) in the oral and anal swab samples of intranasally and orotracheally inoculated bats at steady levels until 7 dpi before waning until 11 dpi (Figure 4A,B). In contrast, we could only detect trace amounts of gRNA in one of the orally inoculated bats at 5 dpi in one oral swab (Figure 4A) and at 7 dpi in one anal swab (Figure 4B), both of which significantly differed from both gRNA levels determined for the other groups (Appendix A). Detectable sgRNA in the samples of intranasally and orotracheally inoculated bats almost reached the same levels as the gRNA, indicating a robust virus replication in the sampled epithelia. However, no sgRNA at all could be detected in the swabs from orally inoculated bats (Figure 4A,B). While at 1 dpi, orotracheally infected fruit bats shed higher levels of virus RNA orally than the intranasally infected animals; this was inversed by 3 dpi when the oral swabs of the intranasally challenged animals displayed distinctly higher levels of gRNA and sgRNA (Figure 4A). In the following days, viral RNA levels were comparable in both groups (Figure 3A), while, with the only exception of a single positive oral swab (10^1.75^ TCID_50_/mL) collected at 3 dpi from an intranasally inoculated bat, no replication-competent virus was detectable (Figure 4A). Fecal shedding of viral RNA (both gRNA and sgRNA) followed the same pattern as the reported oral shedding. Again, despite high levels of viral RNA, no replication-competent virus was detectable, except for one sample at 5 dpi (10^2.75^ TCID_50_/mL) from an orotracheally inoculated bat (Figure 4B).

In our experimental setup, nasal lavage samples generally yield higher levels of viral RNA and replication-competent virus than swab samples [52]. Accordingly, animals inoculated via the intranasal and orotracheal routes shed high levels of viral gRNA of up to 10^7^ genome copies per µL sample, as opposed to a maximum of 10^3^ genome copies in the oral swab samples and slightly lower levels of sgRNA (Figure 4C). This time, we were able to detect viral gRNA, but not sgRNA, also in the orally inoculated animals, albeit at significantly lower levels than in both intranasally and orotracheally inoculated bats (Figure 4C, Appendix A). Moreover, while it was possible to detect the replication-competent virus in intranasally and orotracheally inoculated bats reaching levels of up to 10^4.5^ TCID_50_/mL at 2 and 4 dpi, no replication-competent virus could be detected from orally inoculated bats in any of the nasal lavage samples (Figure 4C).

In the second experiment, we compared the virus-shedding patterns during the early phase of the infection until 6 dpi upon intranasal and oral inoculation with a higher infectious dose (Figure 5). Viral gRNA and sgRNA were detected at similar levels as in the first study in oral (Figure 5A) and anal (Figure 5B) swab samples of the intranasally inoculated bats. However, while no viral RNA (gRNA and sgRNA) could be detected in the oral swab samples of orally inoculated bats (Figure 5A), gRNA shedding was detected at 2 and 5 dpi in anal swabs from the orally inoculated bats, though at significantly lower levels (Figure 5B, Appendix A). Low levels of replication-competent virus were detectable in oral (10^1.75^ to 10^2^ TCID_50_/mL) and anal (10^1.75^ TCID_50_/mL) swabs of intranasally infected bats at 1 and 2 dpi, while no replicating virus was detectable in the orally challenged bats (Figure 5A,B).

Following the challenge with the higher infection dose, viral gRNA was also detected in the nasal lavage samples of orally inoculated bats at 2 and 4 dpi, although at significantly lower levels in comparison to intranasally inoculated bats (Appendix A). At 4 dpi, we detected trace amounts of sgRNA in an orally inoculated bat, again at significantly lower levels compared to the intranasally inoculated bats, where sgRNA was detected at 2 and 4 dpi. Replication-competent virus of up to 10^4.25^ TCID_50_/mL was only detectable in nasal lavage samples from the intranasally inoculated bats (Figure 5C, Appendix A).

#### 3.3.2. Markedly Decreased Tissue Involvement upon Oral Challenge

Upon infection with the low dose, we detected viral gRNA at levels up to 10^6^ copies/µL RNA in the nasal conchae and at levels below 10^3^ copies/µL RNA in all other analyzed tissues of the respiratory and digestive tract, as well as in the brain and spleen in at least one of the intranasally infected animals at 4 dpi (Figure 6A). The replication-competent virus was present in the nasal conchae samples of 2 out of 3 intranasally inoculated bats with titers up to 10^5^ TCID_50_/mL (Figure 6A). IHC confirmed the presence of viral antigen in the respiratory ciliated epithelium of 2 out of 3 nasal conchae samples from intranasally infected bats (Figure 7), while all other tissues were negative using this approach. In orotracheally inoculated bats, gRNA could be detected in all organ samples except the small intestine, brain, and spleen (Figure 6A), while no viral antigen was detected by IHC in these animals. At 14 dpi, viral gRNA was still detectable in the nasal conchae (2/3) of intranasally infected bats, while in orotracheally inoculated bats, gRNA was present in the nasal conchae. Interestingly, we also detected viral gRNA and sgRNA in the brain (1/3, ID FH 15) and in the spleen (ID FH 14) of orotracheally infected bats after challenge with the low dose (Figure 6B). No replication-competent virus was detected at 14 dpi in any of the samples (Figure 6B), and none of the tissues were positive by IHC. Meanwhile, only trace amounts of gRNA were detected in the lung (1/3, ID FH 18) and in the brain (1/3; ID FH 16) of orally challenged bats at 4 dpi at significantly lower levels compared to those in the intranasally inoculated bat samples (Appendix A). Beyond that, we found no signs of virus replication in any tissues of the orally infected animals using PCR, histopathology, and IHC at 4 or 14 dpi. Furthermore, we did not detect viral RNA in the serum samples at 4 or 14 dpi (Figure 6A,B).

Inoculation with the high dose resulted in a similar distribution pattern in intranasally infected animals at 2 dpi (Figure 6C), with higher gRNA and sgRNA levels than determined in the first study at 4 dpi (Figure 6A), which was also confirmed by IHC (Figure 7). Furthermore, using this dose, we detected gRNA and sgRNA in the nasal conchae of one single orally inoculated bat (1/3; ID FH 31) sacrificed at 2 dpi at significantly lower levels compared to the intranasally inoculated bat samples (Appendix A). We also detected gRNA (but no sgRNA) in the trachea, lung lymph node, brain, and mesenteric lymph node of this animal at levels that were significantly lower compared to the intranasally inoculated bat samples (Appendix A). We also detected gRNA in the colon sample of another orally challenged bat (ID FH 33) at levels comparable to the samples from intranasally inoculated bats (Figure 6C). While replication-competent virus could be detected in nasal conchae samples (10^3.75^ to 10^7^ TCID_50_/mL) from intranasally inoculated bats (3/3), no replication-competent virus could be detected in any of the orally inoculated bat samples (Figure 6C).

Notably, we detected high levels of gRNA and sgRNA as well as a replication-competent virus at 10^3^ TCID_50_ in the brain of one intranasally challenged bat (ID FH 25) with the high dose. Since this result could not be confirmed by IHC, we proceeded with an ISH analysis, which confirmed the presence of viral RNA in the cerebellum of this animal (neurons, meningeal cells, ventricular lining cells) (Figure 8). Interestingly, no viral RNA was detectable in the *lobus olfactorius* of this animal, arguing against the *N. olfactorius* as a possible entrance route in this case. All other analyzed tissues of this animal were negative by ISH (Appendix A). However, in this animal, we detected sgRNA in all analyzed samples except for the lung and spleen. This animal was not pregnant, and no physiological abnormalities were observed. No other samples that were analyzed by ISH to follow-up positive PCR results turned out positive by ISH. However, chromogen precipitation was identified in the intestinal lumen of animals that were infected with the higher dose, irrespective of the inoculation route (Appendix A).

At 6 dpi, viral gRNA and sgRNA were detectable in the nasal conchae of all intranasally inoculated bats (3/3). gRNA could be detected in colon (1/3; ID FH 29), small intestine (3/3), lung and mesenteric lymph nodes (both 2/3), and spleen (1/3; ID FH 30) samples. Viral gRNA, but no sgRNA, could be detected in a colon sample of a single (1/3; ID FH 34) orally inoculated bat, as in the first experiment, with no statistically significant difference to the gRNA levels of intranasally inoculated bats (Figure 6D, Appendix A). No replication-competent virus could be isolated at 6 dpi from any of these organ samples (Figure 6D), but IHC revealed one positive cell in the nasal respiratory epithelium (cell not specified due to degenerative changes) of one of these animals. We detected gRNA, but no sgRNA, in the serum sample of an intranasally inoculated bat (1/3; ID FH 27) at 2 dpi (Figure 6C) and in two intranasally inoculated bats (2/3; ID FH 28 and 29) at 6 dpi (Figure 6D). The serum samples of orally inoculated bats were negative for viral RNA at 2 dpi (Figure 6C). However, we detected low levels of gRNA in an orally inoculated bat (1/2; ID FH 36) at 6 dpi (Figure 6D).

No differences were noticeable in the virus-shedding pattern or organ involvement between pregnant and non-pregnant animals in both studies.

### 3.4. Weak Interferon and Antibody Response

#### 3.4.1. Immune Gene Expression After SARS-CoV-2 Challenge Using Different Infection Routes

To further elucidate the mechanism behind the differences in virus shedding and virus distribution related to the infection routes, we investigated expression levels of the type I, II, and III interferons. Although SARS-CoV-2 RNA was detected in most organs upon intranasal and orotracheal infection, the expression of type I, II, and III interferons was not elevated in relation to the uninfected control group, with the exception of a noticeable induction of *IFNβ1* in bronchial lymph nodes upon orotracheal administration of SARS-CoV-2 (Figure 9A). Consistently, the expression of the interferon-inducible chemokine-*CXCL10* in the bronchial lymph node was also highly induced upon orotracheal infection (Figure 9B). Interestingly, orotracheal infection also induced the expression of *IFNγ* in the trachea at 4 dpi (Figure 9C). In contrast, none of the infection routes caused the expression of type III interferon IFNγ3 (Figure 9D).

In the second trial using the high dose, we detected the induction of *IFNβ1* expression at 2 dpi only in the colon of orally infected bats (Appendix A). Although *IFNβ1* expression in the nose was not increased after intranasal infection, *CXCL10* expression was significantly higher upon intranasal infection at 6 dpi (Appendix A). No statistically significant induction of type II or III interferons was observed for any of the infection routes.

#### 3.4.2. Serum Antibodies Against SARS-CoV-2 Only Detectable in Intranasally and Orotracheally Challenged Bats

First, a microsphere-based assay was performed to evaluate the levels of anti-SARS-CoV-2 specific IgG antibodies. As expected, no seroconversion was observed until 6 dpi, while at 14 dpi, all three intranasally inoculated bats and 2 out of 3 orotracheally inoculated bats had seroconverted (Appendix A). None of the orally challenged bats showed seroconversion at any time. In the next step, all sera were re-tested using an indirect ELISA, and a compelling agreement between both methods was shown (Appendix A).

### 3.5. SARS-CoV-2 Virions Are Unstable After Oral Ingestion

#### 3.5.1. Varying pH in Compartments of the *R. aegyptiacus* GIT

First, we measured the pH of the epithelial lining in the *R. aegyptiacus* GIT. The pH measurements showed that the stomach was acidic, with a pH of approx. 3 (±0.5), while all the remaining assayed epithelial linings were mildly acidic with a pH of approx. 6 (Appendix A).

#### 3.5.2. SARS-CoV-2 Is Sensitive to Pepsin Digestion in *R. aegyptiacus* Stomach

To explore the reason why the oral challenges were largely inefficient in establishing a productive SARS-CoV-2 infection, we assessed the stability of SARS-CoV-2 virions in the *R. aegyptiacus* GIT, with the stomach passage being the first significant hurdle for the virus to overcome following oral ingestion. To simulate pepsinogen/pepsin release and activity in the stomach, we incubated stomach homogenates with SARS-CoV-2, using small intestine homogenate as a control. Based on the measured pH of the epithelial lining of the GIT, we incubated the samples with either H_2_O or HCl (final pH~2.5) to activate the pepsinogen. Only the stomach homogenate that was activated with HCl, but not H_2_O, displayed a distinct decrease in protein band intensity (Figure 10A), significantly decreasing by ~69% (±9%) and ~77% (±6%) at 1 and 5 min post activation, which was not observed in the small intestine homogenates (Figure 10B).

We postulate that pepsinogen, via its acid-activated form of pepsin, is present in the stomach homogenate and mediated by proteolysis. Western blot analysis of these lysates confirmed the presence of pepsinogen only in the stomach homogenate but not in the small intestine homogenate. Further, the pepsinogen-to-pepsin conversion was only observed in the HCl-treated fractions (Figure 10C). Western blot analysis showed protein levels significantly decreased by ~85% (±9%) and ~91% (±8%) for the viral S protein, by ~85% (±4%) and ~84% (±12%) for the N protein, and by ~99.9% (±0.1%) for GAPDH at 1 and 5 min post activation respectively, in the samples containing activated pepsin (Figure 10D–F). Further, this reduction was only observed in the stomach homogenates but not in the small intestine homogenates.

#### 3.5.3. Sensitivity of SARS-CoV-2 to Pepsin Digestion

Next, we wanted to confirm that *R. aegyptiacus* pepsin was involved in proteolysis and mediated the SARS-CoV-2 degradation. Pepstatin A, a pepsin and aspartyl protease inhibitor, or the carrier vehicle, DMSO, were therefore added to *R. aegyptiacus* stomach homogenates that contained protease inhibitors without aspartyl protease inhibiting activity. When SARS-CoV-2 was added to the fractions, which were subsequently activated by either H_2_O or HCl, DMSO-treated and HCl-activated lanes showed distinct decreases in protein band intensity, significantly decreasing by ∼68% (±3%), ~81% (±4%), and ~82% (±4%) at 1, 5 and 10 min post activation, respectively (Figure 11A,B). Conversely, samples treated with Pepstatin A and activated with HCl showed no significant change in lane intensity (Figure 11B). Western blot analysis showed that viral S protein levels significantly decreased by ∼90% (±7%), ~99% (±1%), and ~99% (±2%) after 1, 5 and 10 min, and N protein levels significantly decreased by ~71% (±14%), ~77% (±11%), and ~78% (±11%) at 1, 5 and 10 min, respectively, post activation (Figure 10C–E), and cellular GAPDH protein levels significantly decreased by ~99.5% (±0.5%) at 1, 5 and 10 min, post activation (Figure 11C,F). Conversely, the presence of Pepstatin A inhibited this proteolysis in nearly all samples activated with HCl, except for an approx. ~9% (±6%) decrease in the cellular GAPDH protein levels after 10 min (Figure 11F), possibly due to acid hydrolysis [77]. Overall, this shows that the observed decreases were due to the activity of aspartyl proteases, which were inhibited in the Pepstatin A treated samples and required low pH to activate.

#### 3.5.4. SARS-CoV-2 Sensitivity to HCl in *R. aegyptiacus* Stomach

Next, we examined the impact of acidic pH alone, which we had found to be present in the stomach of *R. aegyptiacus* (Appendix A), on the infectivity of SARS-CoV-2 virions. We, therefore, exposed virions to a ~pH 2.5, as used in the pepsin experiments. After only a 1 min exposure, followed by acid neutralization, the virus infectivity drastically decreased by approx. 4 Log_10_, as measured by TCID_50_/mL, and dropped below the detection threshold of 10^1.5^ TCID_50_/mL after 5 min of exposure (Figure 10G). Thus, both pepsin and the low pH in the stomach of *R. aegyptiacus* may contribute to the degradation and inactivation of infectious virions.

## 4. Discussion

Based on the general GIT tropism of coronaviruses in bats and on studies showing that small intestine epithelium from *Rhinolophus spp*. and *Artibeus jamaicensis* bats expressed ACE2 and TMPRSS2 and were susceptible to SARS-CoV-2 infection [36,40] we aimed to analyze the efficiency of an oral SARS-CoV-2 infection in *R. aegyptiacus* fruit bats. Our previous intranasal challenge study using the same bat species had shown a transient infection of the upper respiratory tract with an inefficient transmission to contact animals [34]. Given the increased susceptibility of Golden Syrian hamsters to an orotracheal SARS-CoV-2 infection in comparison to intranasal challenge [52], we also included this route to assess the susceptibility of the lower respiratory tract of this bat species to SARS-CoV-2.

We confirmed an abundant expression of both ACE2 and, to a lesser extent, TMPRSS2 in the analyzed tissues of mock-infected *R. aegyptiacus* bats, confirming an earlier detection of ACE2 expression in the nasal turbinates in this species [78]. The significance of the expression of ACE2 and TMPRSS2 was highlighted in a recent study demonstrating that the distribution of these factors in the respiratory tract correlated with SARS-CoV-2 tissue tropism. The elevated co-expression of ACE2 and TMPRSS2 mRNA has been associated with a higher susceptibility to SARS-CoV-2 infection in some species; for other species, this was only a partial determinant, indicating that other host factors are involved in these species to facilitate permissiveness [79]. While the detection of sgRNA as a surrogate of viral replication was in good correlation with the tissue distribution and abundance of *ACE2* and *TMPRSS2* mRNA in most tissues, we did not determine an abundant expression of both factors in the brain of *R. aegyptiacus*. This result is surprising, given the detection of replication-competent virus in the brain of one intranasally challenged bat, and it supports the hypothesis of other host factors possibly co-influencing the permissiveness of SARS-CoV-2 infection [79]. Meanwhile, we largely exclude the absence of ACE2 and TMPRSS2 expression as the cause for the lack of virus replication in GIT tissues. Moreover, the inconsistent outcome of SARS-CoV-2 infection using intranasal or oral routes is unlikely to be exclusively caused by differences in the ability of SARS-CoV-2 to enter cells in the respiratory and digestive tract but may be due to a yet unknown cofactor, which needs further research.

With the exception of the animals challenged orally and orotracheally with the low dose, all animals gained weight, which is most likely due to the sudden movement restriction in the cage as compared to the aviary where the animals were kept before. Since the animals did not show an initial weight loss in the second study using a higher challenge dose, a cause unrelated to the challenge itself must be assumed for the puzzling transient weight loss observed in the first study. As shown by the temperature and locomotion activity data (Figure 1 and Figure 2), the intranasally challenged animals developed a slight but not significant increase in body temperature and a decrease in locomotion activity, which was not repeated in the second study. Given the overall small group sizes, which is inevitable when working with exotic animals, these observations are most probably due to biological variation. The body temperature values measured during this study fell within the circadian variation of between 34 °C and 41.5 °C as previously reported for members of this same breeding colony [80].

We applied the doubly labelled water (DLW) method to determine the DEE of *R. aegyptiacus* bats within the first 48 h post SARS-CoV-2 infection. This method has so far mostly been applied in field studies to determine the field metabolic rate [81], and we have used it on SARS-CoV-2 infected hamsters [82]. Thus, we decided to also apply this method in this study in order to improve our understanding of the metabolic consequences of a SARS-CoV-2 infection in *R. aegyptiacus* bats, which in past challenge experiments using influenza virus and SARS-CoV-2 did not elicit any clinical signs of infection [34,83]. Although we did observe clear differences in the virological parameters between the groups inoculated using different inoculation routes and with different SARS-CoV-2 doses, the DLW method did not detect any significant differences in the DEE or TWI measurements among the groups, irrespective of the infection route and the infection dose. We were, however, able to show that the DEE determined in this study fits very well within the overall DEE pattern reported for 11 other bat species (Appendix A) [84,85,86,87] and, therefore, conclude that our values are valid. This finding supports the perception of bats being generally more resilient to viral infections than other mammalian species. However, the data show a higher variability in the infected versus the uninfected bats, which may correlate to the overall lower locomotor activity that was determined using activity loggers.

In the 10^6^ TCID_50_ challenge experiment, the orally challenged bats again displayed a lower locomotor activity profile during the night and higher locomotor activity during the day, as opposed to the intranasally challenged bats that showed the lowest locomotor activity in both phases compared to both the uninfected and orally challenged bats. Clearly, the orally challenged bats responded to the infection in a specific way that was distinct from the respiratory-challenged bats, which warrants further study. In this study, the body temperature profiles of all bats remained largely stable, with lower body temperatures during the day and higher body temperatures during the night, which differed slightly from the lower dose experiment, where the orally challenged bats had the lowest body temperature during the night. Again, no elevated body temperature as compared to the physiological variation reported earlier [80] was observed in any of the challenged bats, with the body temperatures falling within the circadian variation previously published for this breeding colony [80]. We observed a slightly increased body temperature to locomotor activity ratio during the night phase in virus-challenged bats compared to the uninfected bats, where we observed a paired increase and decrease of both parameters. It has previously been reported that locomotor activity, especially during flight, increases body temperature in bats [88], indicating that these are inherently linked. Thus, observing this relationship weakened in the virus-challenged bats provided a novel insight that warrants further study to determine how a higher body temperature could be associated with lower locomotor activity, which has already been observed in an earlier study of the same species [89].

As some of the bats used in our study were unexpectedly pregnant, we wanted to elucidate whether this altered physiological and immunological status during the challenge modified the bats’ responses to infection. It has been reported from different field studies that the antibody level, as well as the shedding of henipaviruses and filoviruses, was considerably higher in pregnant bats [90,91,92,93]. In our study, we did not observe such pronounced effects, though, in pregnant animals, a slightly increased shedding of gRNA could be observed. It has been found that pregnancy can affect the immunological status of the mother in mammals [94]. Alternatively, the origin of the animals from our captive breeding colony could also play a role as the animals do not suffer from any food shortage or exceptional stress, rendering them more resilient to stressors like pregnancy or virus infection.

Using the lower challenge dose of 10^4^ TCID_50_, intranasal and orotracheal challenge induced a productive infection and the shedding of replication-competent virus. Conversely, *R. aegyptiacus* bats proved not to be viably infected after oral challenge, as only low levels of gRNA and no sgRNA or replication-competent virus were shed by these animals. Thus, despite the general tropism of Betacoronaviruses for the GIT, which was confirmed by the susceptibility of *Rhinolophus* intestinal epithelia [36], an oral challenge of *R. aegyptiacus* bats turned out not to be effective. While intranasal and orotracheal challenge, but not oral challenge, resulted in robust fecal shedding of viral RNA, it is unlikely that the presence of viral particles in the GIT of these bats was due to the swallowing of infectious respiratory mucus secretions. Given the detection of both gRNA and sgRNA in the whole GIT (small intestine, colon, and mesenteric lymph nodes), these replication sites have more likely been seeded through the systemic dissemination of the virus from the respiratory system, which is supported by the detection of low levels of viral gRNA in the serum of some of the animals challenged with the higher infection dose. Future studies should be designed to decipher the precise routing of the virus to the GIT.

Using the higher dose of 10^6^ TCID_50_ SARS-CoV-2, the intranasal challenge route again induced a productive infection with viable virus reaching the respiratory and digestive tract in the orally challenged bats, as shown by the detection of this gRNA and sgRNA. Also, anal and nasal shedding of viral RNA was observed at low levels from 2 to 5 dpi, which may again be due to an accidental instillation of minute amounts of virus inoculum into the respiratory tract during the oral inoculation or ingestion of infected secretions during the first days after oral challenge.

The intriguing detection of SARS-CoV-2 in the brain tissue of one intranasally challenged animal that was sacrificed at 2 dpi (FH 25) using three independent approaches (RT-PCR, virus titration, and ISH) may indicate hematogenous dissemination. Nevertheless, it should be interpreted with caution and requires confirmation by follow-up studies to identify the route into the brain. While the olfactory bulb has been postulated as a regular point of entry for SARS-CoV-2 virions into the brains of Syrian hamsters [95], the nasal dissemination route is less likely in the case of this bat FH 25, given that the olfactory bulb was free of any detectable SARS-CoV-2 related viral RNA. Microvascular damage and an impairment of the blood-brain barrier have been regularly reported in the highly artificial K18-ACE2 transgenic mouse model for SARS-CoV-2 infections [96], and a breakdown of the Blood Brain Barrier followed by dissemination into the CNS has also been postulated for both severe human cases and for K18-ACE2 transgenic mice by others [96]. In the present case, the involvement of the spleen, as confirmed by RT-PCR analysis, also argues for the hematogenous dissemination.

Based on our findings, we postulate the following infection routes in *R. aegyptiacus* bats: Intranasal infection is most efficient, as it is instilled into the most susceptible tissues of the upper respiratory tract, where a limited virus propagation occurs, followed by generalized virus dissemination which may even lead to the infection of the CNS. Orotracheal instillation into the lower respiratory tract is distinctly less efficient, which is in contrast to what we determined for Golden Syrian hamsters in earlier studies [52], but the virus may still disseminate into other organ systems, including the digestive tract. While both intranasal and orotracheal challenges led to the detection of gRNA and sgRNA in the nasal epithelium, the replication-competent virus could only be detected following the intranasal challenge. Moreover, in these bats, viral RNA, but no replication-competent virus, was detectable in the trachea (both gRNA and sgRNA) and lungs (only gRNA and only at 4 dpi). This indicates a strong tropism of SARS-CoV-2 for the upper respiratory tract in this species, although it has been shown that ACE2 is abundantly expressed in both the upper and lower respiratory tract in *R. aegyptiacus* as well as in the stomach and intestine [78]. In contrast, *R. aegyptiacus* bats are not susceptible to oral SARS-CoV-2 inoculation.

A recent study in Jamaican fruit bats (*A. jamaicensis*) revealed a low susceptibility of lung tissue following an intranasal SARS-CoV-2 infection but a temporary infection of the small intestine. The ineffective lung infection was attributed to low ACE2 expression levels, which was successfully overcome by the transduction of lung cells with human ACE2 [97]. Amino acid sequence alignments of ACE2 orthologs indicate an ACE2 amino acid percentage identity relative to human ACE2 of 78.76% and 79.5% for *R. aegyptiacus* bat and *A. jamaicensis*, respectively. However, when comparing the ACE2 amino acids that directly contact the SARS-CoV-2-RBD [98], the percentage identity shifts to 83.33% and 75% for *R. aegyptiacus* bat and *A. jamaicensis*, respectively. These differences may contribute to the differences in permissiveness and susceptibility between *R. aegyptiacus* bat and *A. jamaicensis*.

Moreover, a recent study examining the susceptibility of *A. jamaicensis* intestinal organoids expressing ACE2 to SARS-CoV-2 infection [40] supports the susceptibility of the small intestine to virus infection. This study found that these organoids could be infected, with limited detection of viral gRNA and sgRNA, but no infectious virus, indicating a stalled replication. This limited infection was attributed to the organoids mounting a successful antiviral IFN response alongside the activation of protective and regenerative pathways [40]. These and other published observations [13,23,24,25,26,91] indicate that CoVs in bats appear to have a tropism for GIT tissues and fecal virus shedding. However, our results indicate that a respiratory infection followed by a systemic dissemination leads to GIT tissue infection and fecal shedding, as the oral ingestion of the virus seems to be an unlikely mode of transmission.

Further, in accordance with a previous study using intestinal organoids from *A. jamaicensis* [40] in the 10^6^ TCID_50_ challenge but not the lower 10^4^ TCID_50_ dose experiment, our analysis of the bat immune response detected the induction of IFNβ1 in the colon of orally challenged bats at 2 dpi, which was also the only organ tissue where we detected viral RNA (gRNA and sgRNA). In contrast, the expression of IFN-I in intestinal organoids from *R. sinicus* bats was not induced by SARS-CoV-2 infection [36]. This discrepancy may be due to different experimental conditions (e.g., infection dose) or to the analyzed bat species. Intriguingly, orotracheal infection with a low dose triggered the induction of IFNβ1 and CXCL10 expression in the pulmonary lymph nodes, indicating the spread of SARS-CoV-2 from the trachea to these lymph nodes. Interestingly, while we determined similar viral RNA levels in the colon of intranasally challenged bats, these did not show a similar induction of IFNβ1. It might be speculated that in the 10^6^ TCID_50_ challenge experiment, the induction of IFNβ1 in the orally inoculated bat colon could have caused fatigue caused by interferon signaling [99], leading to lower locomotor activity. However, given how distinct and unique bat immune responses are as compared to other mammals, such as limiting the expression of inflammatory cytokines following RNA virus infection [100], this may not be the case.

To address the question of why an oral inoculation did not result in a productive infection, we determined the pH of the stomach and intestine of *R. aegyptiacus* bats. Previous work had found differences in the physiology of bat stomachs, highlighting significant differences between insectivore and frugivore bats, with the latter possessing a greater number of pepsinogen-producing chief cells and acid-producing parietal cells [41,42]. Mechanistically, the aspartyl protease pepsin digests ingested protein in the bat’s stomach [101]. It is initially synthesized and secreted into the lumen of the stomach as an inactive zymogen, pepsinogen [102]. In the presence of HCl (and the associated pH decreases to levels below 5), pepsinogen converts to its active form, pepsin. Pepsins are optimally active at pH 1.8 to 3.5, reversibly inactivated at pH 5, and irreversibly denatured at pH 7 [103].

Our work showed that in the frugivore *R. aegyptiacus* bats, the stomach is also acidic, and the pepsinogen-pepsin conversion and activity are dependent on the acidic environment. Our pH measurements in the *R. aegyptiacus* GIT fall within the pH range published for another frugivore bat, *C. perspicillata* [104]. Further, we showed that SARS-CoV-2 virions were susceptible to pepsin-mediated proteolytic degradation. We also showed that a low pH alone was capable of significantly reducing virion infectivity. Hence, pepsin-mediated protein degradation and acidic conditions could cumulatively have diminished infectivity. Thus, upon challenge with a low dose of 10^4^ TCID_50_, the acidic environment and pepsin-mediated degradation were able to block infection, and even the challenge with a high dose of 10^6^ TCID_50_ was not sufficiently high to compellingly overcome this barrier. However, this dose was sufficiently high to allow a transient epithelial infection in the colon, as shown by the detection of gRNA and sgRNA.

One limitation of this study is the low animal number per group, which is inevitable when working with exotic animals under high containment conditions. This partly hampered the statistical analysis, especially of the physiological data (body temperature and activity), but it still allowed us to see certain trends that can be used as a basis for future studies.

## 5. Conclusions

In conclusion, we found that *R. aegyptiacus* bats are permissive to SARS-CoV-2 virus infection following intranasal challenge with a dose as low as 10^4^ TCID_50_, with a dominant viral tropism for the upper respiratory tract. Conversely, we found that *R. aegyptiacus* bats are not susceptible to oral SARS-CoV-2 challenge, and we propose that this was mediated through the stomach as a barrier, which could, however, be partly overcome by a higher infection dose. We also found that upon intranasal or orotracheal inoculation, *R. aegyptiacus* bats may harbor very low levels of replication-competent SARS-CoV-2 virus in their oral cavity and in anal swabs after dissemination from the respiratory tract. Although infected animals may, therefore, theoretically pose a risk for virus transmission to livestock, companion animals, or humans, the transient nature of the infection and the low level of virus shedding reported by [34] and in this report, *R. aegyptiacus* fruit bats are not likely to constitute a viable natural reservoir for SARS-CoV-2 in the field should they become infected. Furthermore, our study approach, addressing the question of SARS-CoV-2 infection in *R. aegyptiacus* bats by examining virus shedding and tissue distribution, physiological reaction, immune response as well as metabolic processes, permitted us to detect the hidden reaction of the animals to an oral challenge that would have easily gone completely unnoticed otherwise. This broad methodological approach may serve as a model for the investigation of other virus infections in bats, where only a transient infection is observed, and the details of the virus dissemination processes are unknown.

## Figures and Tables

**Figure 1 viruses-16-01717-f001:**
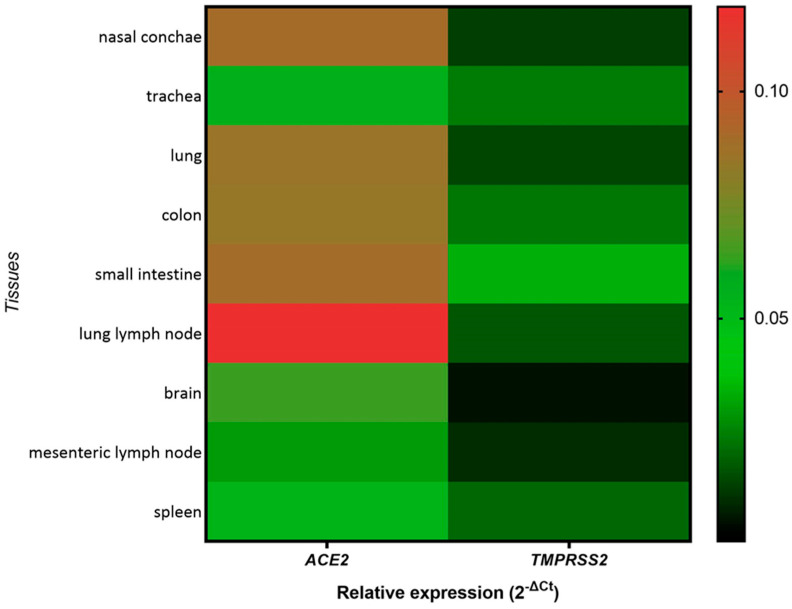
ACE2 and TMPRSS2 relative expression in various tissues of *R. aegyptiacus*. ACE2 and TMPRSS2 expression levels were normalized to the expression of the housekeeping gene EEF1A1. The heatmap shows the means of their relative expression levels from 3 uninfected animals. The color scale represents the relative gene expression from lower (green) to higher levels (red).

**Figure 2 viruses-16-01717-f002:**
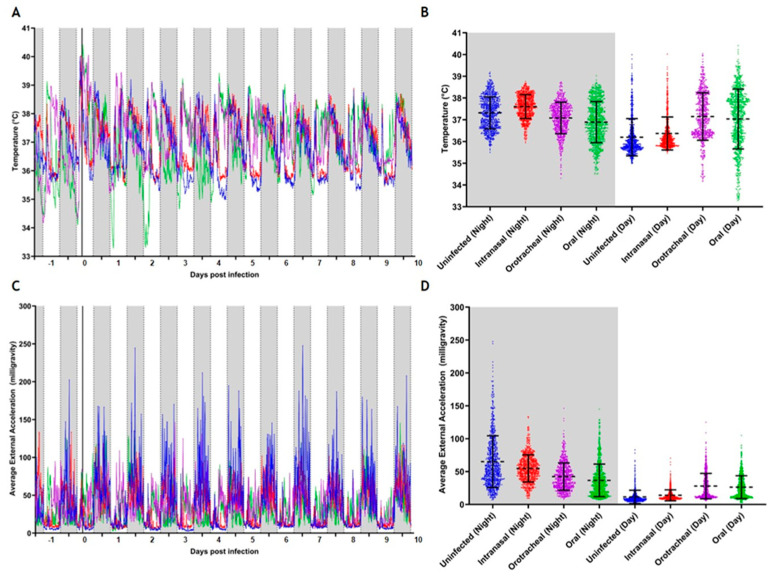
Body core temperature and locomotor activity before, during, and after inoculation with 10^4^ TCID_50_ SARS-CoV-2 via different routes. The data depict uninfected (blue; *n* = 1), intranasally infected (red; *n* = 2), orotracheally infected (purple; *n* = 2), and orally infected (green; *n* = 2) bats, comparing core body temperature (°C) and locomotor activity (measured as average external acceleration, AvgEA, in milli-g). A black line marks the time-point of infection at 0 dpi. Day (white) and night (grey) phases are indicated, with ticks (**A**,**C**) on the x-axis denoting midnight. (**A**) shows the mean body core temperature, and (**C**) shows the mean AvgEA, both as 10 min averages from −1 to 10 dpi. (**B**) illustrates the distribution of body temperature measurements from (**A**), and (**D**) shows the AvgEA from (**C**), both plotted around the mean (dotted black horizontal line) with standard deviation (error bars).

**Figure 3 viruses-16-01717-f003:**
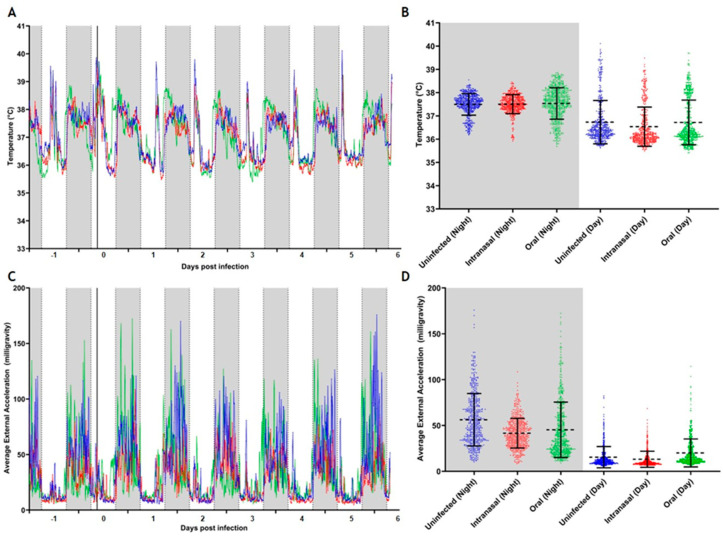
Body core temperature and locomotor activity before, during, and after inoculation with 10^6^ TCID_50_ SARS-CoV-2 via different routes. The data depict uninfected (blue), intranasally infected (red), and orally infected (green) bats, comparing core body temperature (°C) and locomotor activity (measured as average external acceleration, AvgEA, in milli-g). A black line marks the time-point of infection on 0 dpi. Day (white) and night (grey) phases are indicated, with ticks (**A**,**C**) on the *x*-axis denoting midnight. (**A**) shows the mean body core temperature, and (**C**) shows AvgEA, both as 10 min averages from −1 to 6 dpi. (**B**) illustrates the distribution of body temperature measurements from (**A**), and (**D**) shows AvgEA from (**C**), both plotted around the mean (dotted black horizontal line). Mean body core temperature is based on at least two logger-implanted bats per condition (*n* = 2).

**Figure 4 viruses-16-01717-f004:**
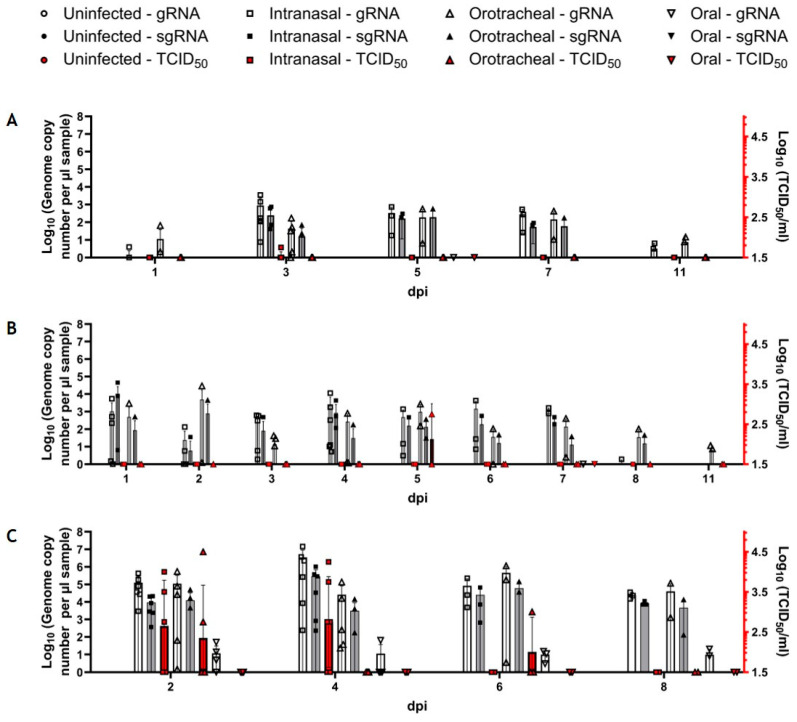
Virus shedding in oral and anal swabs and nasal lavage samples after inoculation of 10^4^ TCID_50_ SARS-CoV-2 (low dose) via different routes. gRNA and sgRNA copy numbers (Log_10_) of viral RNA detected in oral swabs (**A**), anal swabs (**B**), and nasal lavage samples (**C**) as well as replication-competent virus (Log_10_ (TCID_50_/mL)) (red) upon challenge with 10^4^ TCID_50_ using different routes. Note: Circles representing uninfected do not appear on this logarithmic scale, as these values were zero.

**Figure 5 viruses-16-01717-f005:**
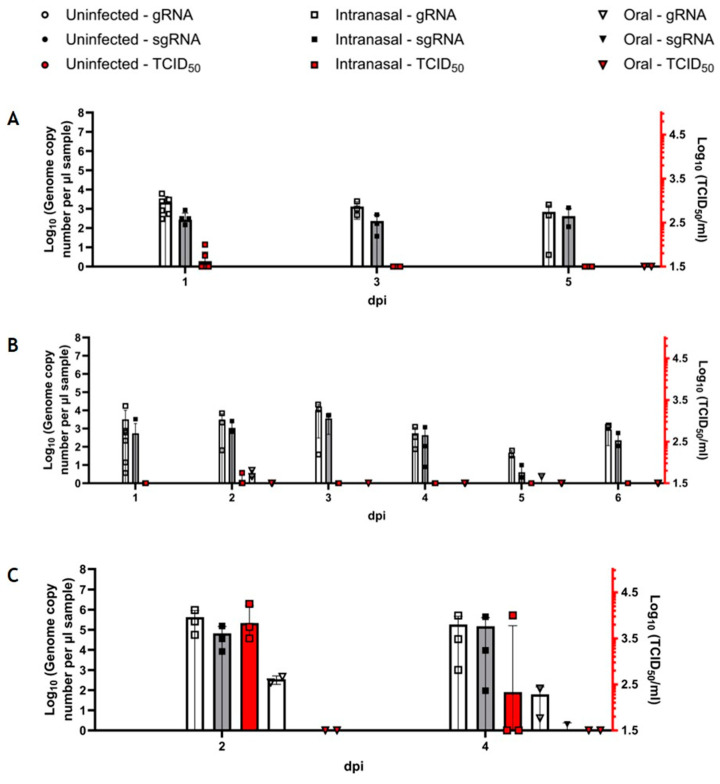
Virus shedding in oral and anal swabs and nasal lavage samples after oral and intranasal inoculation of 10^6^ TCID_50_ SARS-CoV-2 (high dose). gRNA and sgRNA copy numbers (Log_10_) of viral RNA detected in oral swabs (**A**), anal swabs (**B**), and nasal lavage samples (**C**), as well as replication-competent virus (Log_10_ (TCID_50_/mL)) (red) upon challenge with 10^6^ TCID_50_ using different routes. Note: Circles representing uninfected do not appear on this logarithmic scale, as these values were zero.

**Figure 6 viruses-16-01717-f006:**
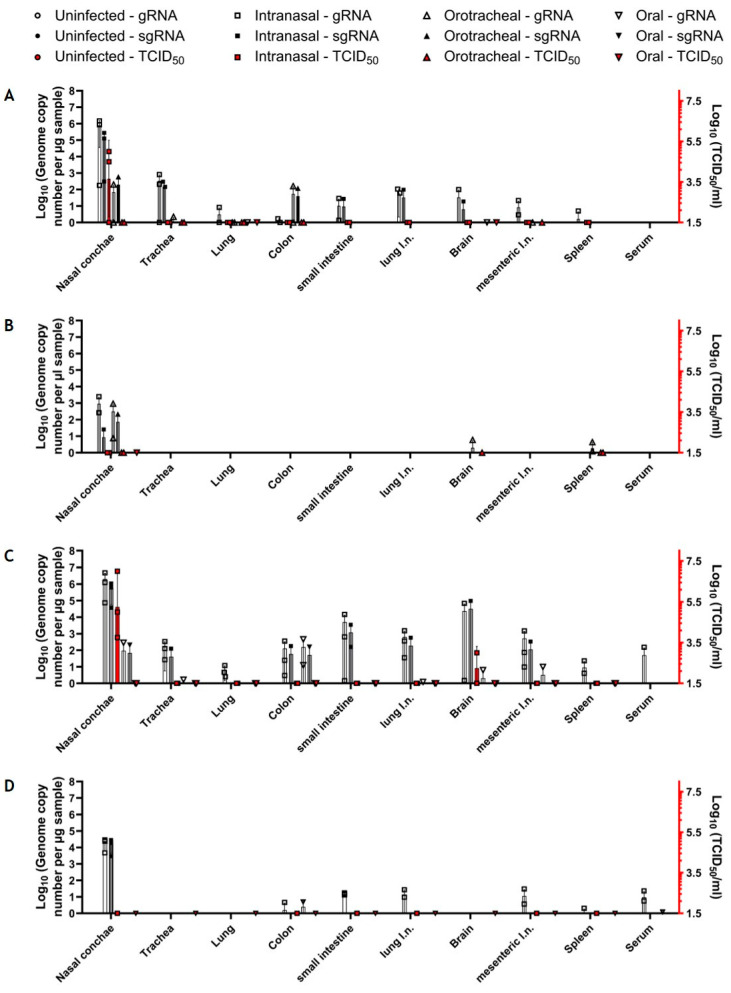
gRNA and sgRNA copy numbers (Log_10_) of SARS-CoV-2 RNA in tissue and serum samples. Inoculation with 10^4^ TCID_50_ at 4 dpi (**A**) and 14 dpi (**B**), and after inoculation with 10^6^ TCID_50_ at 2 dpi (**C**) and 6 dpi (**D**), as well as replication-competent virus (Log_10_ (TCID_50_/mL)) (red). Note: Circles representing uninfected do not appear on this logarithmic scale, as these values were zero.

**Figure 7 viruses-16-01717-f007:**
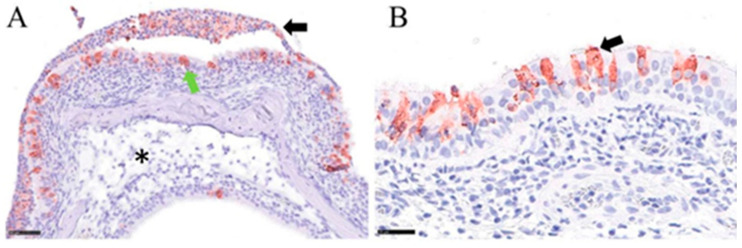
SARS-CoV-2 virus antigen detection in the nasal conchae at 2 and 4 dpi. Nasal respiratory epithelium is immunopositive (green arrow) for Corona-nucleocapsid antigen 2 and 4 days after intranasal infection. Note the intraluminal cellular debris (black arrow) and mucosal edema (black asterisk) 4 dpi, indicating acute, necrotizing rhinitis (**A**). The main target cell is the ciliated (black arrow) respiratory epithelium, shown here at 2 dpi (**B**). Immunohistochemistry, Avidin-Biotin Complex method, aminoethyl carbazole chromogen (red-brown), Mayer’s hematoxylin counterstain (blue). Bar 50 µm (**A**) or 25 µm (**B**).

**Figure 8 viruses-16-01717-f008:**
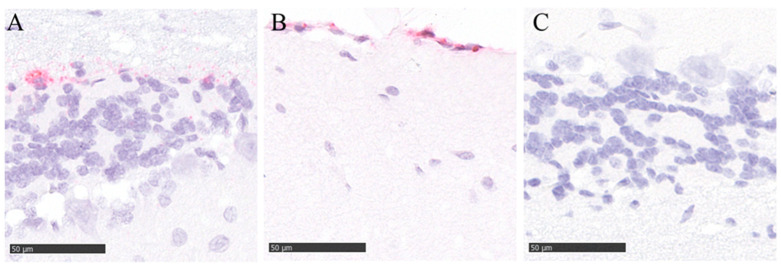
In Situ, Hybridization of cerebellar brain samples from animal ID FH25 necropsied 2 days after intranasal challenge with 10^6^ TCID_50_ SARS-CoV-2. Target cells showing chromogenic labeling include cerebellar neurons (**A**) and meningeal (**B**) cells using probes against the nucleocapsid protein but not with a negative control (**C**) probe (dihydrodipicolinate reductase). RNAscope© in situ hybridization, chromogenic labeling (fast red), Mayer’s hematoxylin counterstain (blue). Bar 50 µm.

**Figure 9 viruses-16-01717-f009:**
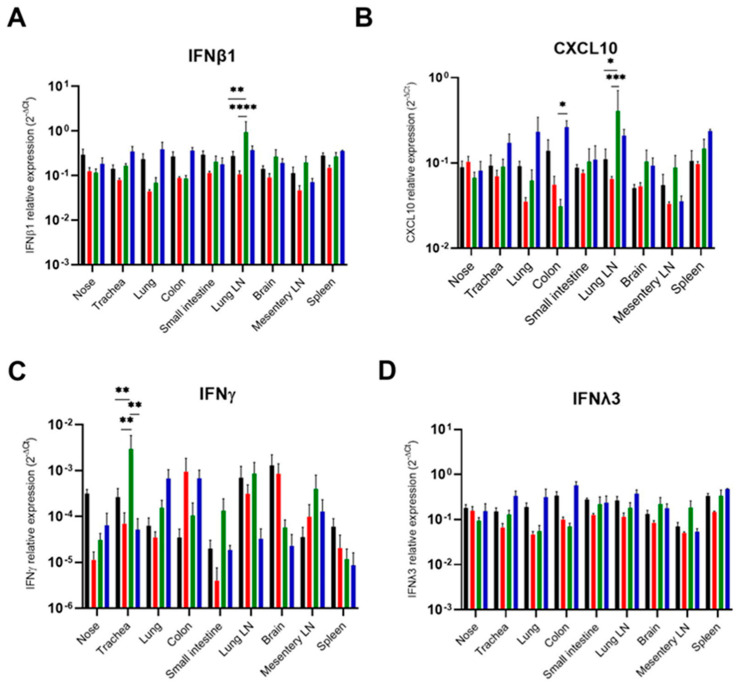
Immune gene expression in intranasally, orotracheally, or orally challenged *R. aegyptiacus* bats. Three Egyptian fruit bats/group were challenged intranasally (red), orotracheally (green), or orally (blue) with a low SARS-CoV-2 dose (10^4^ TCID_50_), while control animals were mock-infected with cell culture medium (black). At 4 dpi, the expression of IFNβ1 (**A**), CXCL10 (**B**), IFNγ (**C**), and IFNλ3 (**D**) was analyzed by qPCR. Data are shown as mean ± SEM; *p*-values were calculated using two-way ANOVA with Tukey’s multiple comparison test. (*) *p* < 0.05, (**) *p* < 0.01, (***) *p* < 0.001, (****) *p* < 0.0001.

**Figure 10 viruses-16-01717-f010:**
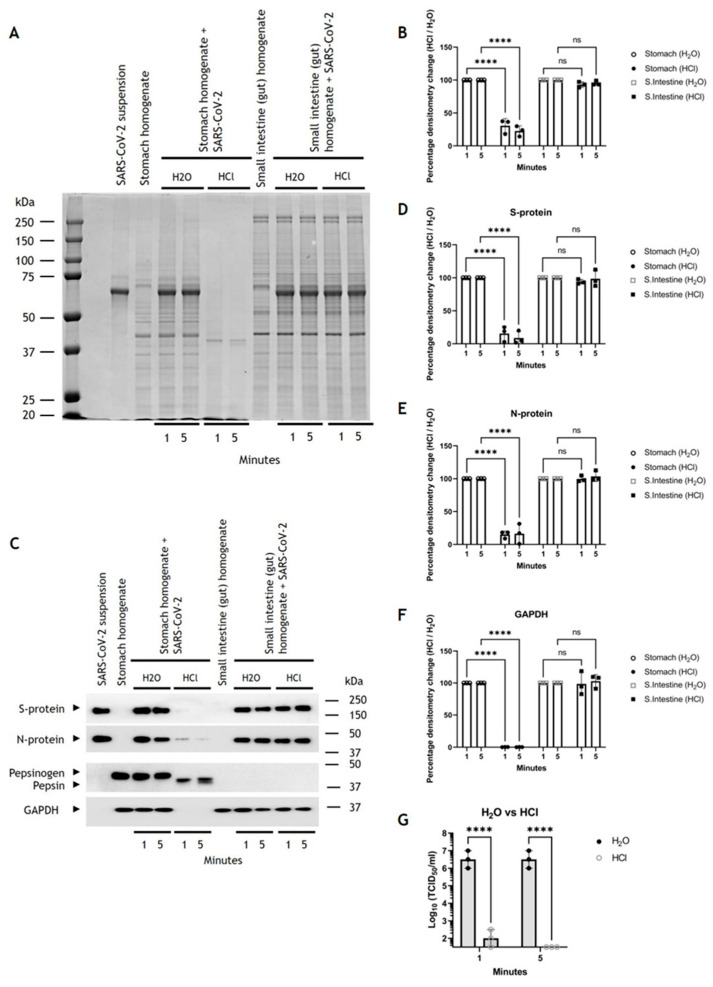
*R. aegyptiacus* stomach homogenate, but not small intestine homogenate, activated with HCl, effectively digests SARS-CoV-2 S and N proteins. Coomassie-stained SDS-PAGE gel (**A**) and densitometry analysis of lanes in (**B**) normalized to the corresponding H_2_O-treated sample. Representative Western blots of SDS-PAGE gels for SARS-CoV-2 spike- and nucleocapsid protein, pepsin, and GAPDH (**C**). Densitometry analysis of Western blots normalized to the corresponding H_2_O treated sample (**D**–**F**). HCl decreases SARS-CoV-2 infectivity (**G**). Error bars denote the standard deviation (±SD) of three independent experiments (*n* = 3); *p*-values were calculated using two-way ANOVA with post-hoc Tukey HSD Test. Not significant (ns), (****) *p* < 0.0001.

**Figure 11 viruses-16-01717-f011:**
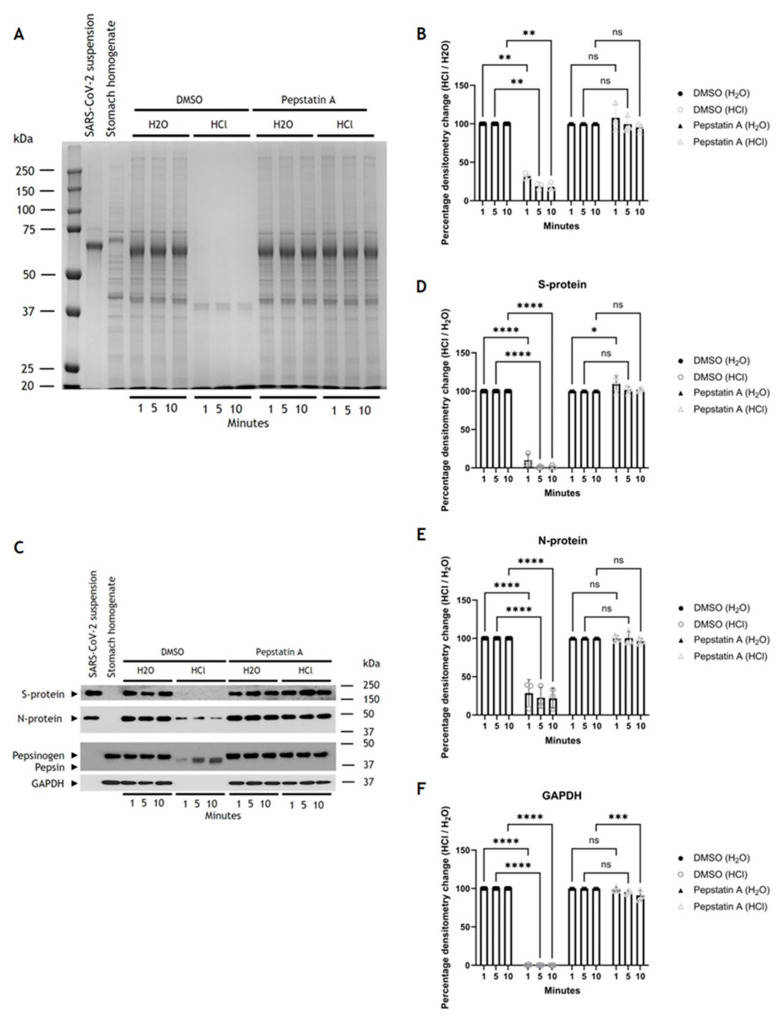
*R. aegyptiacus* stomach homogenate activated with HCl digests SARS-CoV-2 virions, but pepsin proteolysis is inhibited by Pepstatin A. Coomassie-stained SDS-PAGE gel (**A**), and densitometry analysis of lanes in (**B**), normalized to the corresponding H_2_O treated sample. Representative Western blots of SDS-PAGE gels for SARS-CoV-2 spike- and nucleocapsid protein, pepsinogen/pepsin, and GAPDH (**C**). Densitometry analysis of Western blots normalized to the corresponding H_2_O treated sample (**D**–**F**). Error bars denote the standard deviation (±SD) of three independent experiments (*n* = 3); *p*-values were calculated using two-way ANOVA with post-hoc Tukey HSD Test. Not significant (ns), (*) *p* < 0.05, (**) *p* < 0.01, (***) *p* < 0.001, (****) *p* < 0.0001.

**Table 1 viruses-16-01717-t001:** Primer sequences are used for the quantification of immune gene expression.

Target	Primer Identification	Sequence
*EEF1A1*	EEF1A1 F	5′-GTATGCCTGGGTCTTGGATAAA-3′
	EEF1A1 R	5′-GCCTGTGATGTGCCTGTAA-3′
*IFNβ*	IFNβ F	5′-CAGAAGGAGGACGCAGTATT-3′
	IFNβ R	5′-GGCTGTATCCAGAAGGTCTATC-3′
*IFNγ*	IFNγ F	5′-GGTTTGGGTGATTTTGGGTTCTTC-3′
	IFNγ R	5′-CACTGCTTTGAATGGTCGGGTTAT-3′
*IFNλ3*	IFNλ3 F	5′-GGCTTTGGAGGCTGAACT-3′
	IFNλ3 R	5′-AGGCGGAAGAGGTTGAATG-3
*CXCL10*	CXCL10 F	5′-CTTTAGAACTACACGCTGTGTCTGC-3′
	CXCL10 R	5′-ACCTTTCCTTGCTAATTGCTTTCAGT-3′
*ACE2*	ACE2 F	5′-TATTGAGCCAACACTGGGAAC-3′
	ACE2 R	5′-CGACAAAGATGAGCAGGACAA-3′
*TMPRSS2*	TMPRSS2 F	5′-GGTCACTTTGAAGAACAGCATC-3′
	TMPRSS2 R	5′-TCATTTGTCGGTAGATCCAGTC-3′

## Data Availability

The original data presented in the study are openly available in Zenodo at: 10.5281/zenodo.14013011.

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
