# Peer review of "Increased Susceptibility of Rousettus aegyptiacus Bats to Respiratory SARS-CoV-2 Challenge Despite Its Distinct Tropism for Gut Epithelia in Bats"

_viruses, 2024, doi:10.3390/v16111717_

Round 1

Reviewer 1 Report

Comments and Suggestions for Authors

32 - would specify that this is fecal shedding of RNA

Materials and Methods - first level headings (117, 125, 131, etc) were in plain text and subheadings were italicized - not sure if this is due to journal standards, but with so many components to this study would also be nice to visually distinguish the headings

155 - might leave mentioning the incidental pregnancies til the results section

259 - define RT as room temperature the first time used

Results section - mostly, but not entirely in the same order as the methods, would be nice to have them entirely mirrored

390 - add "the weight of" intranasally infected bats remailed unchanged "during this time period"

405 - curious if the bats in all groups were sampled at a consistent time of day, and if that might have affected the locomotion and temperature data?

Figures 4 and 5 - would explicitly call "low dose" and "high dose" in addition to TCID50 value for clarity's sake

Appreciated the comparison of genomic and subgenomic RNA with the viral culture data - nicely done.

497 - is this sentence referring to your previous work?

Figures 4 and 6 - these two figures are hard to read as busy and small as they are - would be nice to have bigger

Comments on the Quality of English Language

79 - replace "is resulting" with "resulted"

203 - necropsy, not autopsy 

397 - this sentence does not make sense as written, missing a word, I imagine something like "observed" after the word predominantly 

Author Response

Reviewer 1:

32 - would specify that this is fecal shedding of RNA

Autor response: We have amended the text and RNA shedding has now been specified.

Materials and Methods - first level headings (117, 125, 131, etc) were in plain text and subheadings were italicized - not sure if this is due to journal standards, but with so many components to this study would also be nice to visually distinguish the headings

Author response: Thank you to the reviewer for pointing this out, the first level headings have now been underlined for better demarcation.

155 - might leave mentioning the incidental pregnancies til the results section

Autor response: Thank you for the suggestion, we have moved the relevant section from materials and methods to the results section.

259 - define RT as room temperature the first time used

Author response: Done.

Results section - mostly, but not entirely in the same order as the methods, would be nice to have them entirely mirrored

Autor response: Thank you, while we strove to maintain the order of both methods and results, this was not always possible, for example, the Figure 1 data presenting the relative expression of ACE2 and TMPRSS2 and the Figure 9 data presenting the immune gene expression in various tissues of R. aegyptiacus are both based on the methods shown under “Analysis of early and late immune response” in the materials and methods section. As the preponderance of genes analyzed encompassed immune genes, these methods were located further back in the materials and methods section, reflecting the position of the immune gene expression in the results section.

390 - add "the weight of" intranasally infected bats remained unchanged "during this time period"

Author response: Thank you, we have amended the sentence.

405 - curious if the bats in all groups were sampled at a consistent time of day, and if that might have affected the locomotion and temperature data?

Author response: The bats in all groups were sampled at a consistent time in the morning, always in the same order. Indeed, we did observe a body temperature spike, paired with low locomotor activity during the animal handling when samples were taken.

Figures 4 and 5 - would explicitly call "low dose" and "high dose" in addition to TCID50 value for clarity's sake.

Author response: We have now included "low dose" and "high dose" in the respective figure legends after the TCID50 value.

Appreciated the comparison of genomic and subgenomic RNA with the viral culture data - nicely done.

Author response: Thank you very much!

497 - is this sentence referring to your previous work?

Author response: Yes, we apologize for missing this in the first version. A reference has now been added.

Figures 4 and 6 - these two figures are hard to read as busy and small as they are - would be nice to have bigger.

Author response: Thank you for the suggestion, we have amended the figures and increased the font size for these figures to improve readability.

Comments on the Quality of English Language

79 - replace "is resulting" with "resulted"

Author response: Done.

203 - necropsy, not autopsy

Author response: This has been corrected.

397 - this sentence does not make sense as written, missing a word, I imagine something like "observed" after the word predominantly

Author response: Thank you for highlighting this, we have amended the sentence.

Reviewer 2 Report

Comments and Suggestions for Authors

The paper investigated nasal, orotracheal and nasal inoculation routes and subsequent infection dynamics of SARS-CoV2 in experimentally infected Rousettus bats. The findings are interesting and certainly answer many questions specific to SARS-CoV2 infection in Rousettus bats. The conclusions are sound and may be extrapolated to other bat coronaviruses to a limited degree. It is still infection of a human-adapted virus in a bat species not known to be a reservoir for that virus diversity – sarbecoviruses, and so overall conclusions regarding bat coronavirus infection modes within ‘bats’ should be made with caution (such as in the abstract: “These findings change our general understanding of the pathophysiology of coronavirus infections in bats” – should rather be: “These findings may influence our general understanding of the pathophysiology of coronavirus infections in bats”. At present it is not known if a similar infection mode would be seen if Rousettus were to be infected with one of its own host coronaviruses, or if Rhinolophus would be infected with SARS-CoV2 or a related bat sarbecovirus. The findings highlighting the issue of stomach pH is compelling toward understanding how poor oral infection routes may be to viral infection – and thus require further exploration of natural infection of coronaviruses among bats. Lastly, I think the authors should do a bit more literature searching for papers on coronavirus detection among bat oral swabs – which have been much lower than from GIT samples. Nasal swabs are very infrequently collected.

Specific points to highlight:

In the abstract (and throughout the paper) I don’t think it is necessary to use a symbol to denote the alpha and beta in alpha- and betacoronaviruses – just write it out. If you really want to use a symbol you can use it for abbreviated form of AlphaCoV/BetaCoV after it has been defined for the first time.

Intro:

Lines 47-48: Interestingly, bats were not known to harbour coronaviruses (CoV) before the SARS-CoV epidemic in 2003 [13-16]. = perhaps just clarify this statement that it was not known that bats harbour coronaviruses before SARS-CoV due to limited/lacking surveillance in bats.

Lines:57-58: Rhinolophus bats are the likely reservoir hosts for SARS-CoV [29], and they have also been postulated as a reservoir host for SARS-CoV-2 [7]. = It would be better to say that Rhinolophus bats are the reservoir hosts for the diversity of Sarbecoviruses (Sarbecovirus subgenus within the Betacoronavirus genus) that include both SARS-CoV [29] and SARS-CoV-2 [7].

Methods:

are well described (just correct Tab.1 to Table 1 on line 246). There are however a lot of methods and some figures would make some of the steps easier to follow (just a suggestion)– like the sets of experimental bats or the treatments to evaluate SARS-CoV-2 sensitivity to pH and pepsin.

Results:

Lines397-398:  Needs some clarity: “We predominantly nocturnal circadian rhythm with body temperature and activity levels 397 being highest during the night in both experiments”

Discussion:

Lines 732: it should be Rhinolophus spp. for plural species within the genus and not Rhinolophus ssp. (which denotes subspecies for which a species needs to be given)

Lines 755-757: I think more evidence is required for the statement like: Moreover, the inconsistent outcome of SARS-CoV-2 infection using intranasal or oral routes is unlikely caused by differences in the ability of SARS-CoV-2 to enter cells in the respiratory and digestive tract. 

Author Response

The paper investigated nasal, orotracheal and nasal inoculation routes and subsequent infection dynamics of SARS-CoV2 in experimentally infected Rousettus bats. The findings are interesting and certainly answer many questions specific to SARS-CoV2 infection in Rousettus bats. The conclusions are sound and may be extrapolated to other bat coronaviruses to a limited degree. It is still infection of a human-adapted virus in a bat species not known to be a reservoir for that virus diversity – sarbecoviruses, and so overall conclusions regarding bat coronavirus infection modes within ‘bats’ should be made with caution (such as in the abstract: “These findings change our general understanding of the pathophysiology of coronavirus infections in bats” – should rather be: “These findings may influence our general understanding of the pathophysiology of coronavirus infections in bats”.

Author response: We thank the reviewer for highlighting this, we have amended the sentence in the abstract.

At present it is not known if a similar infection mode would be seen if Rousettus were to be infected with one of its own host coronaviruses, or if Rhinolophus would be infected with SARS-CoV2 or a related bat sarbecovirus. The findings highlighting the issue of stomach pH is compelling toward understanding how poor oral infection routes may be to viral infection – and thus require further exploration of natural infection of coronaviruses among bats.

Lastly, I think the authors should do a bit more literature searching for papers on coronavirus detection among bat oral swabs – which have been much lower than from GIT samples. Nasal swabs are very infrequently collected.

Author response: we have specifically addressed coronavirus detection in oral swab field samples in the introduction now, that yielded a lower detection rate compared to fecal/anal swab samples.

Specific points to highlight:

In the abstract (and throughout the paper) I don’t think it is necessary to use a symbol to denote the alpha and beta in alpha- and betacoronaviruses – just write it out. If you really want to use a symbol you can use it for abbreviated form of AlphaCoV/BetaCoV after it has been defined for the first time.

Author response: Thank you for the suggestion. The symbols have now been replaced for Alpha- and Betacoronaviruses.

Intro:

Lines 47-48: Interestingly, bats were not known to harbour coronaviruses (CoV) before the SARS-CoV epidemic in 2003 [13-16]. = perhaps just clarify this statement that it was not known that bats harbour coronaviruses before SARS-CoV due to limited/lacking surveillance in bats.

Author response:The text has been amended by adding “presumably due to limited surveillance in bats”.

Lines:57-58: Rhinolophus bats are the likely reservoir hosts for SARS-CoV [29], and they have also been postulated as a reservoir host for SARS-CoV-2 [7]. = It would be better to say that Rhinolophus bats are the reservoir hosts for the diversity of Sarbecoviruses (Sarbecovirus subgenus within the Betacoronavirus genus) that include both SARS-CoV [29] and SARS-CoV-2 [7].

Author response: The sentence has been amended accordingly.

Methods:

are well described (just correct Tab.1 to Table 1 on line 246). There are however a lot of methods and some figures would make some of the steps easier to follow (just a suggestion)– like the sets of experimental bats or the treatments to evaluate SARS-CoV-2 sensitivity to pH and pepsin.

Author response: Thank you for the suggestion. We have corrected this. Additionally, we have included a new supplementary figure (Figure S 7) depicting the experimental outline of both high and low dose experiments, including sample taking and necropsy time points.

Results:

Lines397-398:  Needs some clarity: “We predominantly nocturnal circadian rhythm with body temperature and activity levels being highest during the night in both experiments”

Author response: We have clarified this sentence by adding ‘observed a’.

Discussion:

Lines 732: it should be Rhinolophus spp. for plural species within the genus and not Rhinolophus ssp. (which denotes subspecies for which a species needs to be given)

Author response: Thank you for pointing this out and apologies for this error, we have corrected this.

Lines 755-757: I think more evidence is required for the statement like: Moreover, the inconsistent outcome of SARS-CoV-2 infection using intranasal or oral routes is unlikely caused by differences in the ability of SARS-CoV-2 to enter cells in the respiratory and digestive tract.

Author response: We have amended and ameliorated the sentence as follows to emphasize that is more of a hypothesis: “Moreover, the inconsistent outcome of SARS-CoV-2 infection using intranasal or oral routes is unlikely to be exclusively caused by differences in the ability of SARS-CoV-2 to enter cells in the respiratory and digestive tract, but may be due to a yet unknown cofactor, which needs further research.”